# Pruning Aggregation Parameters for Large Language Models

## Abstract

Pruning is a highly effective approach for compressing large language models (LLMs). By strategically reducing model size, pruning significantly decreases both latency and GPU memory usage during inference, resulting in more efficient and cost-effective deployment of these models. Despite their effectiveness, current structured pruning algorithms have limitations. They still require extensive continued pre-training on large datasets to achieve model compression. Moreover, most of these methods are unable to reduce the memory usage of the key-value cache during generation tasks. In this work, we propose a novel pruning algorithm that requires no additional training and targets specific parameters within LLMs. We classify the model's parameters into three categories: aggregation, transformation, and normalization. Our method primarily focuses on pruning the aggregation parameters in the higher layers of the model. To further improve the performance of the pruned LLM, we also introduce a rescaling parameter that adjusts the output of the pruned block. We conduct comprehensive experiments on a wide range of LLMs, including LLaMA3.1-8B/70B, Qwen2-7B/72B, Gemma2-9B, and Mistral-7B-v0.3. Our evaluation includes both generation and discriminative tasks across various benchmarks. The results consistently demonstrate that our method outperforms recent block pruning methods. This improvement is particularly notable in generation tasks, where our approach significantly outperforms existing baselines.

## 1 Introduction

Large language models (LLMs) (Touvron et al., 2023; OpenAI et al., 2023; Jiang et al., 2023; Yang et al., 2024; Gemma2-Team et al., 2024), pre-trained on extensive text data from across the internet, have achieved remarkable performance in downstream tasks such as information retrieval (Asai et al., 2024), code generation (Guo et al., 2024a), and mathematical reasoning (Wang et al., 2023; Yang et al., 2023; Huang et al., 2024). These LLMs, however, contain a huge number of parameters, resulting in substantially slower inference speed compared to their smaller counterparts. To address this issue in generation tasks, a common approach is to use key-value (KV) cache (Pope et al., 2023), which stores intermediate computation results. While this technique effectively trades space for time, speeding up inference, it also significantly increases GPU memory consumption. As reported in Zhou et al. (2024), the KV cache size can exceed the LLM model size during peak usage, and the inference latency increases as the KV cache size grows. As a result, one major bottleneck for LLM serving is GPU memory consumption.

Recent strategies to improve LLM efficiency primarily fall into two categories. The first category focuses on the models themselves, aiming to reduce inference latency and GPU memory consumption through pruning (Frantar & Alistarh, 2023; Ma et al., 2023; Jaiswal et al., 2023; Xia et al., 2024; Ashkboos et al., 2024; Xu et al., 2024; Jaiswal et al., 2024a; Zhang et al., 2024c; Dong et al., 2024c; Yin et al., 2024a;b; Zhao et al., 2024) or quantizing (Frantar et al., 2023; Xiao et al., 2023; Chee et al., 2023; Lin et al., 2024). The second category targets the KV cache, specifically for generation tasks, by either compressing (Dong et al., 2024b) or quantizing (Zhang et al., 2024d; Liu et al., 2024b) it to decrease GPU memory usage during inference. Among these approaches, structured pruning (Xia et al., 2024) searches for crucial substructures within the model while pruning other substructures through continued pretraining on extensive text datasets. However, a significant limitation of most current pruning algorithms is their inability to reduce the GPU memory consumption within the KV cache. To address this issue, KV cache compression algorithms like LESS (Dong et al., 2024b) have

been proposed, which maintain a constant-size KV cache by generating condensed representations of less important tokens. These approaches (Xia et al., 2024; Dong et al., 2024c), however, typically require designing specific learning objectives and loss functions, followed by extensive retraining of the base model on large text corpora to achieve the desired goal. We argue that these methods require an additional training phase, introducing significant computational overhead. Moreover, these approaches may struggle to maintain performance in domains not well-covered in the extra training data (Xia et al., 2024). This raises an important question: *Can we develop a training-free algorithm that effectively reduces GPU memory consumption with respect to the KV cache?* Our work addresses this challenge by drawing inspiration from an unexpected source: the intriguing connections between Graph Neural Networks (GNNs) (Kipf & Welling, 2016; 2017; Hamilton et al., 2017; Veličković et al., 2019) and LLMs. By exploring the parallels in their computation processes, we uncover insights that lead to a novel, training-free method for improving LLM efficiency.

Recent studies (Joshi, 2020; Ying et al., 2021; Kim et al., 2022; Nguyen et al., 2023; Barbero et al., 2024) have uncovered connections between GNNs and Transformers (Vaswani et al., 2017). The fundamental principle of GNNs is to aggregate information from neighboring nodes, resulting in smooth representations across the graph. This principle finds a parallel in LLMs, where the flow of contextual information can be conceptualized as a GNN operating on a fully connected graph, with connections governed by a causal attention mask. In this conceptualization, the process involves aggregating information from previous tokens to update the representations of subsequent ones. However, this aggregation process is not without challenges. In GNNs, while increasing the number of layers allows for the incorporation of higher-order neighbor information and potentially smoother representations, it also risks over-smoothing (Li et al., 2018). This phenomenon can lead to node representations converging to similar values, ultimately making them indistinguishable from one another. To address this issue in GNNs, GCNII (Chen et al., 2020a) has been developed, utilizing initial residual connections (Huang et al., 2017) and identity mappings, formulated as:

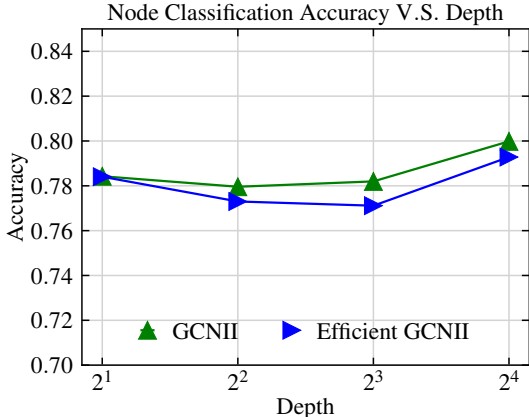

Figure 1: Performance comparison between GCNII and its efficient variant on the Pubmed dataset. The experiment evaluates both models with varying depths ($L$ = 2, 4, 8, and 16). The efficient GCNII demonstrates performance comparable to the original GCNII across all tested depths, despite its reduced computational complexity.

$$\boldsymbol{H}^{(\ell+1)} = \sigma\left(\left((1-\alpha_\ell)\,\tilde{\boldsymbol{\mathcal{A}}}\boldsymbol{H}^{(\ell)} + \alpha_\ell \boldsymbol{H}^{(0)}\right)\left((1-\beta_\ell)\,\boldsymbol{I} + \beta_\ell \boldsymbol{H}^{(\ell)}\right)\right), \tag{1}$$

where $\tilde{\boldsymbol{\mathcal{A}}} = \tilde{\boldsymbol{D}}^{-1/2}\tilde{\boldsymbol{A}}\tilde{\boldsymbol{D}}^{-1/2}$, $\boldsymbol{A}$ is the adjacency matrix, $\tilde{\boldsymbol{A}} = \boldsymbol{A} + \boldsymbol{I}$, $\tilde{\boldsymbol{D}}$ is the degree matrix of $\tilde{\boldsymbol{A}}$, and $\alpha_\ell$, $\beta_\ell$, and $\boldsymbol{W}^{(\ell)}$ are the $\ell$-th layer parameters. Although GCNII addresses over-smoothing, its accuracy improves by only 1.6% when increasing layers from 2 to 16 (Figure 1), at the cost of an eightfold increase in computation. Aggregation in GNNs is particularly computationally expensive, especially in large graphs, accounting for up to 90% of total training and inference time (Liu et al., 2023). GCNII can be made more efficient by reducing the number of aggregation operations during inference while keeping the training process unchanged. This modified version can be formulated as follows:

$$\boldsymbol{H}^{(\ell+1)} = \begin{cases} \sigma\left(\left((1-\alpha_\ell)\,\boxed{\tilde{\boldsymbol{\mathcal{A}}}\boldsymbol{H}^{(\ell)}} + \alpha_\ell \boldsymbol{H}^{(0)}\right)\left((1-\beta_\ell)\,\boldsymbol{I} + \beta_\ell \boldsymbol{W}^{(\ell)}\right)\right) & \text{if } \ell \leq \frac{L}{2}, \\ \sigma\left(\left((1-\alpha_\ell)\,\boxed{\boldsymbol{H}^{(\ell)}} + \alpha_\ell \boldsymbol{H}^{(0)}\right)\left((1-\beta_\ell)\,\boldsymbol{I} + \beta_\ell \boldsymbol{W}^{(\ell)}\right)\right) & \text{if } \ell > \frac{L}{2}, \end{cases} \tag{2}$$

where $L$ is the depth of GCNII. As shown in Figure 1, we achieve comparable performance to GCNII while halving the computational cost of aggregation during inference.

This phenomenon motivates us to explore whether a similar approach could be applied to LLMs. Similar to the computationally expensive aggregation in GNNs, the self-attention module in LLMs poses significant computational challenges. It exhibits quadratic time and memory complexity with

---

**Algorithm 1** LLM inference with KV cache at each layer

---

1: **Input**: Current token's hidden state $\boldsymbol{H}_t \in \mathbb{R}^{1\times d}$, previous keys $\boldsymbol{K}_{\text{cache}} \in \mathbb{R}^{(t-1)\times(h\times d_k)}$, previous values $\boldsymbol{V}_{\text{cache}} \in \mathbb{R}^{(t-1)\times(h\times d_k)}$

2: **Step 1. Self-Attention**:

3: Apply Layer Normalization: $\boldsymbol{H}_t' = \boldsymbol{w}_1 \odot \left( \frac{\boldsymbol{H}_t}{\sqrt{\frac{1}{d}\sum_{i=1}^{d}\left([\boldsymbol{H}_t]_{:,i}\right)^2 + \epsilon}} \right)$, where $\boldsymbol{w}_1 \in \mathbb{R}^d$

4: Apply Linear Projection: $\boldsymbol{Q}_t = \boldsymbol{H}_t'\boldsymbol{W}_Q$, $\boldsymbol{K}_t = \boldsymbol{H}_t'\boldsymbol{W}_K$, $\boldsymbol{V}_t = \boldsymbol{H}_t'\boldsymbol{W}_V$, where $\boldsymbol{W}_Q \in \mathbb{R}^{d\times(h\times g\times d_k)}$, $d = h \times g \times d_k$, and $\boldsymbol{W}_K, \boldsymbol{W}_V \in \mathbb{R}^{d\times(h\times d_k)}$

5: Update KV Cache: $\boldsymbol{K}_{\text{cache}} \in \mathbb{R}^{t\times(h\times d_k)} \leftarrow \text{Concat}\left(\boldsymbol{K}_{\text{cache}}, \boldsymbol{K}_t\right)$, $\boldsymbol{V}_{\text{cache}} \in \mathbb{R}^{t\times(h\times d_k)} \leftarrow \text{Concat}\left(\boldsymbol{V}_{\text{cache}}, \boldsymbol{V}_t\right)$

6: Reshape $\boldsymbol{Q}_t$, $\boldsymbol{K}_{\text{cache}}$, and $\boldsymbol{V}_{\text{cache}}$: $\mathbf{Q} \in \mathbb{R}^{1\times h\times g\times d_k} \leftarrow \text{Reshape}\left(\boldsymbol{Q}_t\right)$, $\mathbf{K} \in \mathbb{R}^{t\times h\times d_k} \leftarrow \text{Reshape}\left(\boldsymbol{K}_{\text{cache}}\right)$, $\mathbf{V} \in \mathbb{R}^{t\times h\times d_k} \leftarrow \text{Reshape}\left(\boldsymbol{V}_{\text{cache}}\right)$

7: Apply Rotary Position Embedding (RoPE): $\mathbf{Q} \leftarrow \text{RoPE}\left(\mathbf{Q}\right)$, $\mathbf{K} \leftarrow \text{RoPE}\left(\mathbf{K}\right)$, where RoPE is parameter-free

8: Repeat $\mathbf{K}$ and $\mathbf{V}$: $\mathbf{K}' \in \mathbb{R}^{t\times h\times g\times d_k} \leftarrow \text{Repeat}\left(\mathbf{K}\right)$, $\mathbf{V}' \in \mathbb{R}^{t\times h\times g\times d_k} \leftarrow \text{Repeat}\left(\mathbf{V}\right)$, where $\mathbf{K}'_{:,h_k,:g,:} = \mathbf{K}_{:,h_k,:}$ and $\mathbf{V}'_{:,h_k,:g,:} = \mathbf{V}_{:,h_k,:}$

9: Compute Attention Scores $\mathbf{A} \in \mathbb{R}^{1\times t\times h\times g}$: $\mathbf{A} = \text{softmax}\left(\frac{\mathbf{Q}\mathbf{K}'^T}{\sqrt{d_k}}\right)$, where $\mathbf{K}'^T \in \mathbb{R}^{t\times h\times d_k\times g}$

10: Aggregate: $\mathbf{H}_{\text{attn}} = \boxed{\mathbf{A}\mathbf{V}'}$ through the second dimension    ▷ Aggregation over the context

11: Reshape $\boldsymbol{H}_{\text{attn}}$: $\boldsymbol{H}_{\text{attn}} \in \mathbb{R}^{1\times d} \leftarrow \text{Reshape}\left(\mathbf{H}_{\text{attn}}\right)$

12: Add Residual Connection: $\boldsymbol{H}_t = \boldsymbol{H}_t + \boldsymbol{H}_{\text{attn}}\boldsymbol{W}_O$, where $\boldsymbol{W}_O \in \mathbb{R}^{d\times d}$

13: **Step 2. Feedforward Network (FFN)**:

14: Apply Layer Normalization: $\boldsymbol{H}_t' = \boldsymbol{w}_2 \odot \left( \frac{\boldsymbol{H}_t}{\sqrt{\frac{1}{d}\sum_{i=1}^{d}\left([\boldsymbol{H}_t]_{:,i}\right)^2 + \epsilon}} \right)$, where $\boldsymbol{w}_2 \in \mathbb{R}^d$

15: Apply FFN: $\boldsymbol{H}_{\text{FFN}} = \left(\sigma\left(\boldsymbol{H}_t'\boldsymbol{W}_{\text{gate}}\right) \odot \left(\boldsymbol{H}_t'\boldsymbol{W}_{\text{up}}\right)\right)\boldsymbol{W}_{\text{down}}$, where $\sigma$ is non-linear activation function, $\boldsymbol{W}_{\text{gate}}, \boldsymbol{W}_{\text{up}} \in \mathbb{R}^{d\times d_i}$, and $\boldsymbol{W}_{\text{down}} \in \mathbb{R}^{d_i\times d}$

16: Add Residual Connection: $\boldsymbol{H}_t = \boldsymbol{H}_t + \boldsymbol{H}_{\text{FFN}}$

17: **Output**: Updated keys $\boldsymbol{K}_{\text{cache}}$, updated values $\boldsymbol{V}_{\text{cache}}$, and updated $\boldsymbol{H}_t$

---

respect to sequence length (Dao et al., 2022), making it a bottleneck in LLM serving. In this work, inspired by the efficient version of GCNII discussed earlier, we propose a training-free pruning strategy called AggregationPruner that targets only the query and key parameters in the higher layers of LLMs. By selectively pruning these aggregation parameters, AggregationPruner compresses the model and achieves a significant reduction in GPU memory consumption associated with the KV cache during generation tasks. Considering the complex and black-box nature of LLMs, our approach carefully avoids pruning transformation or normalization parameters to minimize the potential negative impacts of pruning on downstream tasks. To further enhance performance, we incorporate a rescaling parameter for the output of pruned blocks. Extensive experiments demonstrate that our method outperforms recent block pruning algorithms (Men et al., 2024; Zhong et al., 2024; Gromov et al., 2024; He et al., 2024; Siddiqui et al., 2024; Liu et al., 2024a; Zhang et al., 2024a; Jaiswal et al., 2024b; Chen et al., 2024; Kim et al., 2024) across a wide range of downstream tasks and testing LLMs. Notably, our approach shows significant performance improvement in generation tasks while maintaining the same memory consumption with Self-AttentionPruner and LayerPruner (Gromov et al., 2024; He et al., 2024) during inference.

## 2 PRELIMINARIES

### 2.1 DECODER-ONLY LARGE LANGUAGE MODEL

In decoder-only LLMs, information flows through self-attention modules, with each token aggregating context from all preceding tokens in the sequence. This autoregressive process enables the model to generate each subsequent token based on the information from earlier tokens. To preserve the causal structure of language generation during training, attention is masked, preventing tokens from accessing information from future positions in the sequence. For a given input sentence

Table 1: Classification of LLM layer parameters based on their functional roles.

| Type | $\boldsymbol{w}_1$ | $\boldsymbol{W}_Q$ | $\boldsymbol{W}_K$ | $\boldsymbol{W}_V$ | $\boldsymbol{W}_O$ | $\boldsymbol{w}_2$ | $\boldsymbol{W}_{\text{gate}}$ | $\boldsymbol{W}_{\text{up}}$ | $\boldsymbol{W}_{\text{down}}$ |
|---|---|---|---|---|---|---|---|---|---|
| Aggregation Parameter | | ✓ | ✓ | | | | | | |
| Transformation Parameter | | | | ✓ | ✓ | | ✓ | ✓ | ✓ |
| Normalization Parameter | ✓ | | | | | ✓ | | | |

$\boldsymbol{x} = \{x_1, \ldots, x_n\}$, LLMs employ a standard language modeling objective. This objective aims to maximize the following: (Radford et al., 2018):

$$L(\boldsymbol{x}) = \sum_i \log P\left(x_i \mid x_{i-1}, \cdots, x_1; \Theta\right), \tag{3}$$

where $P\left(x_i \mid x_{i-1}, \cdots, x_1; \Theta\right)$ represents the probability of token $x_i$ given all preceding tokens and the model parameters $\Theta$. Conceptually, LLMs can be viewed as operating on a complete graph structure, with tokens serving as nodes and attention scores as edges. Both LLMs and GNNs employ a similar strategy for information processing: they iteratively refine representations by incorporating contextual information. In LLMs, this context is derived from preceding tokens in a sequence, while in GNNs, it comes from neighboring nodes in a graph. Despite operating in different domains, these two model types share a fundamental approach to information aggregation and propagation. This shared mechanism allows both LLMs and GNNs to generate context-aware representations.

Algorithm 1 illustrates the inference computation process of a decoder-only LLM layer. We categorize the model parameters into three functional groups: aggregation, transformation, and normalization, as detailed in Table 1. Aggregation parameters, such as $\boldsymbol{W}_Q$ and $\boldsymbol{W}_K$, are used to compute attention scores within the adjacency matrix $\boldsymbol{A}$. These parameters enable the model to aggregate information from preceding tokens, integrating context and capturing dependencies among tokens. Transformation parameters, such as $\boldsymbol{W}_V$, $\boldsymbol{W}_O$, and $\boldsymbol{W}_{\text{gate}}$, apply linear transformations and feedforward operations to the hidden states of tokens. These parameters are crucial for the model's ability to process input and generate output. Normalization parameters, like $\boldsymbol{w}_1$ and $\boldsymbol{w}_2$, play a significant role in stabilizing the training process. By maintaining a consistent scale in the output, they help prevent issues such as vanishing or exploding gradients. In this work, we propose a pruning algorithm that specifically targets the aggregation parameters to improve the LLMs' efficiency.

## 2.2 BLOCK PRUNING STRATEGIES

Recent research (Men et al., 2024; Zhong et al., 2024; Gromov et al., 2024; He et al., 2024; Siddiqui et al., 2024) has revealed the presence of redundant parameters in the higher layers of LLMs. These studies demonstrate that selectively pruning certain blocks within these higher layers has little performance degradation on downstream discriminative tasks. These pruning strategies can be classified into three distinct approaches: Self-AttentionPruner, FFNPruner, and LayerPruner. Each targets different components of the model:

- Self-AttentionPruner: This method bypasses the self-attention computation module (Step 1 in Algorithm 1), removing parameters across all three categories: Aggregation, Transformation, and Normalization.
- FFNPruner: By skipping the feed-forward network computation process (Step 2), this approach primarily prunes Transformation and Normalization parameters.
- LayerPruner: This method skips an entire layer, resulting in the removal of all parameter types within that layer.

These studies have introduced heuristic metrics to evaluate the importance of blocks within each layer of LLMs. A consistent finding across these works is the greater significance of parameters in lower layers compared to those in higher layers. Consequently, pruning algorithms typically target parameters in higher layers while preserving those in lower layers.

This phenomenon can be intuitively explained through the lens of GNNs. The fundamental principle of GNNs is to aggregate information from neighboring nodes to achieve smoother representations. However, as the number of GNN layers increases, node representations tend to converge towards a

---

**Algorithm 2** LLM inference with AggregationPruner at each layer

---

1: **Input**: Current token's hidden state $\boldsymbol{H}_t \in \mathbb{R}^{1 \times d}$
2: **Step 1. Transformation without Self-Attention**:
3: Apply Layer Normalization as Line 3 in Algoritm 1
4: Apply Linear Projection: $\boldsymbol{V}_t = \boldsymbol{H}'_t \boldsymbol{W}_V$, where $\boldsymbol{W}_V \in \mathbb{R}^{d \times (h \times d_k)}$
5: Reshape $\boldsymbol{V}_t$: $\mathbf{V} \in \mathbb{R}^{1 \times h \times d_k} \leftarrow \text{Reshape}\left(\boldsymbol{V}_t\right)$
6: Repeat $\mathbf{V}$: $\mathbf{V}' \in \mathbb{R}^{1 \times h \times g \times d_k} \leftarrow \text{Repeat}\left(\mathbf{V}\right)$, where $\mathbf{V}'_{0,h_k,:g,:} = \mathbf{V}_{0,h_k,:}$
7: Reshape $\mathbf{V}'$: $\boldsymbol{V}_t \in \mathbb{R}^{1 \times d} \leftarrow \text{Reshape}\left(\mathbf{V}'\right)$
8: Add Residual Connection: $\boldsymbol{H}_t = \boldsymbol{H}_t + \boxed{\alpha \boldsymbol{V}_t \boldsymbol{W}_O}$          ▷ Introduce a rescaling parameter
9: **Step 2. Feedforward Network (FFN)**:
10: Apply FFN and Residual Connection as Lines 14-16 in Algorithm 1
11: **Output**: Updated $\boldsymbol{H}_t$

---

common value. Beyond a certain point, adding more layers contributes minimally to changing node representations, which can be formulated as follows:

$$\lim_{\ell \to \infty} \left\| \boldsymbol{H}^{(\ell+1)} - \boldsymbol{H}^{(\ell)} \right\|_F^2 = \lim_{\ell \to \infty} \left\| \boldsymbol{P}^{(\ell+1)} \boldsymbol{X} - \boldsymbol{P}^{(\ell)} \boldsymbol{X} \right\|_F^2 = 0. \tag{4}$$

While the propagation matrix $\boldsymbol{P}$ in GNNs is static, the attention matrix in LLMs is dynamic. Despite this difference, recent studies (Shi et al., 2022; Nguyen et al., 2023) have revealed that Transformers can also experience over-smoothing, similar to GNNs. This phenomenon provides insight into the behavior of the Block Importance (BI) metric proposed by Men et al. (2024):

$$\text{BI}^{(\ell)} = 1 - \mathbb{E}_{\boldsymbol{H},t} \frac{\boldsymbol{H}_t^{(\ell)} \cdot \boldsymbol{H}_t^{(\ell+1)}}{\left\| \boldsymbol{H}_t^{(\ell)} \right\|_2 \left\| \boldsymbol{H}_t^{(\ell+1)} \right\|_2}. \tag{5}$$

The BI metric tends to decrease as the layer index $\ell$ increases. This observation explains why recent pruning algorithms target blocks in higher layers: these layers contribute less unique information. Informed by these insights, our work also focuses on pruning aggregation parameters in the higher layers of LLMs.

## 3 AGGREGATIONPRUNER

In this section, we first discuss the motivation behind our proposed AggregationPruner in Section 3.1 and 3.2. Then, we provide the details of our pruning algorithm in Section 3.3.

### 3.1 THE BOTTLENECK IN LLM SERVING

In applications such as chatbots and content generation tools, which handle a high volume of daily API requests, maintaining low latency is crucial. This is typically achieved by batching multiple requests for inference, thereby reducing computational waste. Moreover, modern LLMs employ the KV cache to accelerate inference by storing intermediate results. While effective, this approach leads to increased memory consumption as the number of requests grows. To illustrate the scale of memory consumption from the KV cache, we use an example from PagedAttention (Kwon et al., 2023). A 13B parameter OPT model (Zhang et al., 2022), capable of generating up to 2048 tokens, requires approximately 800 KB of GPU memory per token. This can lead to a potential consumption of 1.6 GB per request. Given that LLM operations are primarily constrained by memory bandwidth (Dao et al., 2022), the amount of memory access becomes the primary factor in determining runtime. Consequently, understanding the mechanism by which LLMs generate and utilize the KV cache is essential for optimizing resource utilization.

### 3.2 DISTINCT ROLES OF PARAMETER TYPES IN LARGE LANGUAGE MODELS

#### 3.2.1 THE ROLE OF AGGREGATION PARAMETER

As previously discussed, aggregation parameters play a crucial role in calculating attention scores, which are essential for aggregating contextual information from preceding tokens to subsequent ones.

---

**Algorithm 3** Top-down $\alpha$ grid search for AggregationPruner

---

1: **Input**: Pruned LLM, number of pruned layers $P$, alpha search range $0, 0.1, 0.2, \cdots, 1.0$
2: **Initialize**: $\alpha_{best} = 1.0^P$ (list of length $P$)
3: **for** $\ell = L$ to $L - P + 1$ step $-1$ **do**                    ▷ Start from the top layer
4:     $\text{PPL}_{\text{best}} = \infty$
5:     **for** $\alpha \in 0, 0.1, 0.2, \cdots, 1.0$ **do**
6:         Set $\alpha_\ell = \alpha$ in the pruned LLM
7:         $\text{PPL} = \text{Perplexity} (\text{LLM}_{\text{pruned}} (\alpha_{best}))$
8:         **if** $\text{PPL} < \text{PPL}_{\text{best}}$ **then**
9:             $\text{PPL}_{\text{best}} = \text{PPL}$
10:           $\alpha_{best}[\ell] = \alpha$
11:         **end if**
12:     **end for**
13: **end for**
14: **Output**: Optimal $\alpha_{best}$ for each pruned layer

---

This process involves computing the inner product of queries and keys, resulting in quadratic time and memory complexity with respect to sequence length. To accelerate the generation of subsequent tokens, modern LLMs typically employ a KV cache mechanism as illustrated in Algorithm 1. This approach stores previously calculated keys and values, thereby reducing computational overhead. When generating a new token, the model only needs to compute the query, key, and value for the last token in the sequence. It then combines the KV cache with the last token's query and key to aggregate information from previous tokens, integrating this context into the last token's representation. By avoiding the need to recompute keys for each token, this approach significantly accelerates the calculation of attention scores. These scores are then used to aggregate contextual information from the V cache and the last token's value. This optimization strategy greatly enhances inference speed by minimizing redundant computations, particularly for long sequence generation tasks.

While the KV cache significantly accelerates inference, it also introduces substantial GPU memory consumption. As previously discussed, higher layers in LLMs typically contribute less unique information to the model's output. Leveraging this insight, our work focuses on pruning aggregation parameters in these higher layers to reduce the size of the KV cache. This approach aims to balance the trade-off between inference speed and memory efficiency, optimizing overall model performance.

### 3.2.2 THE ROLE OF TRANSFORMATION PARAMETERS

Transformation parameters in LLMs comprise two main components: $\boldsymbol{W}_v$ and $\boldsymbol{W}_o$ in the Self-Attention module, and $\boldsymbol{W}_{\text{gate}}, \boldsymbol{W}_{\text{down}}, \boldsymbol{W}_{\text{up}}$ in the Feed-Forward Network (FFN). These parameters apply linear transformations on token embeddings and, as some research (Anderson, 1972; Kohonen, 1972; Geva et al., 2021; Meng et al., 2023) suggests, serve as storage of compressed knowledge (Deletang et al., 2024; Lester et al., 2024) derived from vast internet-scale text data.

Current block pruner methods risk discarding valuable stored knowledge when pruning these transformation parameters. Furthermore, since pruning aggregation parameters already provides substantial memory savings, further pruning of transformation parameters results in diminishing returns. This additional pruning could also introduce potential issues, especially when it comes to maintaining performance across various downstream tasks.

### 3.3 OUR PROPOSED PRUNING ALGORITHM: AGGREGATIONPRUNER

Building on the insights discussed above, we introduce AggregationPruner, a novel pruning algorithm designed specifically for LLMs. This approach strategically focuses on pruning only the aggregation parameters in the higher layers of LLMs, preserving the knowledge-rich transformation parameters. By doing so, AggregationPruner achieves substantial memory efficiency gains while preserving the model's core knowledge base. The computation process for the higher layers, incorporating our pruning strategy, is detailed in Algorithm 2. It's important to note that many LLMs employ Grouped-query attention (GQA) (Ainslie et al., 2023). Therefore, a modification is required to accommodate this architecture as shown in Algorithm 2. Specifically, in Line 6, we must replicate the

---

**Algorithm 4** LLM inference with AggregationPruner

---

1: **Input**: Token sequence $\mathcal{T}$, the number of layers $L$, the number of pruning layers $P$
2: **for** $\ell = 1$ to $L$ **do**
3:     **if** $\ell \leq L - P$ **then**
4:         Run Algorithm 1 on $\mathcal{T}$: Compute $\boldsymbol{H}^{(\ell)}$
5:     **else**
6:         Run Algorithm 2 on $\mathcal{T}$: Compute $\boldsymbol{H}^{(\ell)}$
7:     **end if**
8: **end for**
9: **Output**: Final token representation $\boldsymbol{H}^{(L)}$

---

value matrix $\boldsymbol{V}$ a total of $g - 1$ times, where $g$ represents the number of query groups. This replication ensures compatibility with the GQA mechanism, enabling our AggregationPruner to seamlessly integrate with modern LLM architectures.

When modifying the higher layers of an LLM, we propose that the original residual connection coefficient of 1 may no longer be optimal. Inspired by GCNII, which uses a decreasing coefficient to address the diminishing unique information in higher layers caused by over-smoothing, we introduce a rescaling parameter. This parameter, denoted as $\alpha$, adjusts the pruned block's output within the residual connection, as shown in Line 8 of Algorithm 2.

Determining the optimal value for $\alpha$ presents a challenge. Traditional retraining methods are not applicable due to $\alpha$'s non-differentiable nature. While some recent works have employed Zeroth-Order Optimization (Guo et al., 2024b; Zhang et al., 2024b) to estimate gradients during fine-tuning, we propose a simpler, more efficient approach: a greedy search strategy. Our method involves calculating the perplexity of the pruned LLM to identify the optimal $\alpha$. To simplify the search process, we adopt a top-down approach. We begin by determining $\alpha$ for the uppermost layer and then use this value as a starting point for the subsequent layer. This cascading strategy significantly reduces the search space. The entire process is implemented as a grid search as illustrated in Algorithm 3, balancing efficiency with thoroughness. This approach allows us to fine-tune the rescaling parameter across layers, optimizing the model's performance post-pruning without the need for extensive retraining. The development of more complex search strategies leaves room for future work. The inference process, which incorporates AggregationPruner, is detailed in Algorithm 4.

## 4 EXPERIMENTS

In this section, we present a comprehensive evaluation of our proposed pruning algorithm, assessing its performance across six LLMs and ten diverse benchmarks. By conducting experiments on various LLMs and benchmarks, we aim to establish consistent and reliable results.

### 4.1 SETUP

**Baselines.** We evaluate AggregationPruner against three baselines: FNNPruner, LayerPruner, and Self-AttentionPruner, which are described in Section 2.2. These baseline methods employ various heuristic metrics to determine which layers should be pruned. While there may be minor variations in the specific layers selected for pruning, all these methods generally prune from top to bottom. In this work, we evaluate performance by pruning different blocks within the selected layers. To ensure a fair comparison, we apply a top-to-bottom pruning approach for all methods as shown in Algorithm 4.

**Testing LLMs.** We evaluate all pruning algorithms on 6 LLMs: LLaMA3.1-8B/70B (Touvron et al., 2023), Qwen2-7B/72B (Yang et al., 2024), Gemma2-9B (Gemma2-Team et al., 2024), and Mistral-7B-v0.3 (Jiang et al., 2023). All experiments are conducted using Nvidia H100/A100 80G GPUs. However, due to memory constraints, we are unable to load the full weights of LLaMA3.1-70B and Qwen2-72B directly onto the GPU. To address this limitation, we employ the bnb quantization method provided by Hugging Face to compress these two models to 4-bit precision for our experiments.

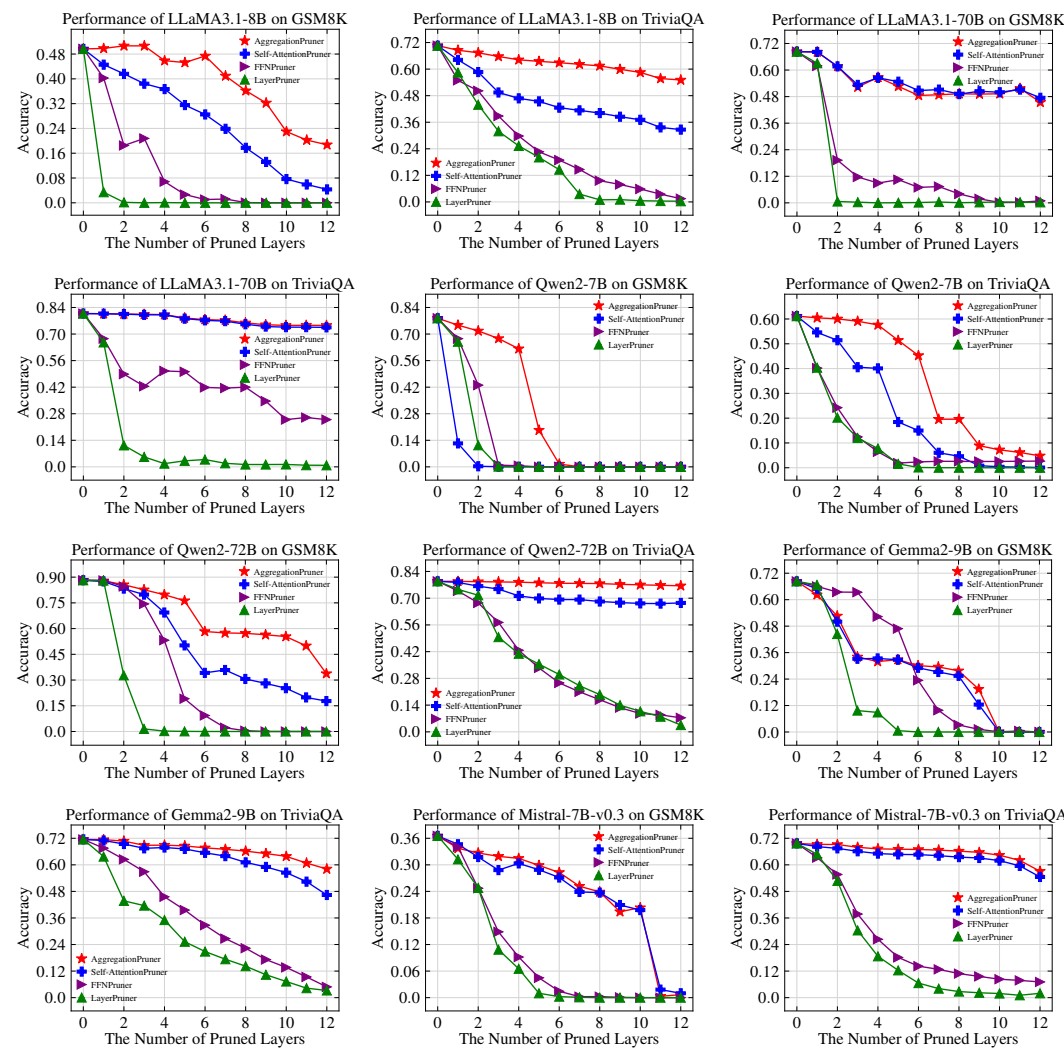

Figure 2: We evaluate the performance of 6 LLMs, including LLaMA3.1-8B/70B, Qwen2-7B/72B, Gemma2-9B, and Mistral-7B-v0.3, on generation tasks such as GSM8K and TriviaQA. Our evaluation involves progressively pruning layers, starting from 0 and extending up to 12 layers.

**Benchmarks.** We employ the `lm-evaluation-harness` package (Gao et al., 2021) to conduct experiments on both generation and discriminative tasks. While current LLM deployments primarily focus on generation tasks, discriminative tasks are often used to evaluate overall model performance. It's important to note that LLMs only generate the KV cache during generation tasks. Discriminative tasks, on the other hand, involve providing inputs and directly obtaining results, such as classification labels or regression values. Our generation tasks include 5-shot GSM8K (Cobbe et al., 2021) and 5-shot TriviaQA (Joshi et al., 2017). For discriminative tasks, we use 7-shot CommonsenseQA (Talmor et al., 2019), 5-shot WinoGrande (Sakaguchi et al., 2019), 25-shot ARC-Challenge (Clark et al., 2018), 0-shot BoolQ (Clark et al., 2019), 0-shot OpenBookQA (Mihaylov et al., 2018), 0-shot PIQA (Bisk et al., 2020), 0-shot MedQA (Jin et al., 2020), and 5-shot MMLU (Hendrycks et al., 2021). We report the accuracy for these tasks as recommended by the `lm-evaluation-harness` package.

## 4.2 RESULTS

**Generation Tasks.** Figure 2 presents the results for all pruning algorithms, showing the superior performance of our proposed AggregationPruner across multiple generation tasks and language models. Our method outperforms the baselines on both generation tasks with LLaMA3.1-8B, Qwen2-

Table 2: The Performance of LLaMA3.1-8B on Discriminative Tasks.

| #Layers | Method | CommonSenseQA | WinoGrande | ARC-Challenge | BoolQ | OpenBookQA | PIQA | MedQA | MMLU | Average |
|---|---|---|---|---|---|---|---|---|---|---|
| 0 | No Pruning | 73.6 | 77.2 | 54.7 | 82.1 | 33.4 | 80.0 | 59.9 | 65.2 | 65.8 |
| 2 | FFNPruner | 73.3 | 68.7 | 48.1 | 81.1 | 35.2 | 76.2 | 59.7 | **65.1** | 63.4 |
| | LayerPruner | 73.4 | 67.2 | 47.5 | 81.1 | **37.6** | 75.6 | 59.9 | **65.1** | 63.4 |
| | Self-AttentionPruner | 71.2 | 77.8 | 50.4 | 75.9 | 31.0 | 78.8 | **60.2** | 62.1 | 63.4 |
| | AggregationPruner | **73.9** | **78.0** | **53.4** | **81.7** | 32.6 | **79.7** | 60.0 | 64.9 | **65.5** |
| 4 | FFNPruner | 73.3 | 66.2 | 45.1 | 77.4 | 32.2 | 75.0 | 53.9 | 62.6 | 60.7 |
| | LayerPruner | 71.7 | 65.7 | 45.3 | **78.1** | **34.0** | 74.1 | 57.3 | 63.4 | 61.2 |
| | Self-AttentionPruner | 71.1 | 76.7 | 49.5 | 56.6 | 30.4 | 77.9 | **60.4** | 61.6 | 60.5 |
| | AggregationPruner | **74.4** | **77.5** | **52.7** | 78.0 | 33.0 | **79.1** | 60.1 | **65.0** | **65.0** |
| 6 | FFNPruner | 71.2 | 65.6 | 41.2 | 71.2 | 30.0 | 71.3 | 46.3 | 56.0 | 56.6 |
| | LayerPruner | 72.4 | 60.9 | 43.9 | **79.3** | **33.4** | 71.9 | 53.5 | 61.3 | 59.6 |
| | Self-AttentionPruner | 71.0 | 77.4 | 50.3 | 52.4 | 30.6 | 77.8 | 59.0 | 62.1 | 60.1 |
| | AggregationPruner | **74.3** | **77.7** | **53.2** | 75.7 | 32.6 | **78.9** | 59.2 | 64.8 | **64.6** |
| 8 | FFNPruner | 72.6 | 64.7 | 37.5 | 62.2 | 27.2 | 68.6 | 55.3 | 62.8 | 56.4 |
| | LayerPruner | 61.9 | 62.3 | 41.0 | 62.3 | 30.4 | 69.8 | 53.2 | 54.5 | 54.4 |
| | Self-AttentionPruner | 71.6 | 76.6 | 49.1 | 51.8 | 30.4 | 77.7 | 58.4 | 62.0 | 59.7 |
| | AggregationPruner | **74.0** | **77.7** | **53.6** | **74.8** | **33.2** | **79.0** | **59.9** | **64.8** | **64.6** |
| 10 | FFNPruner | 71.6 | 63.9 | 32.2 | 62.1 | 24.4 | 65.6 | 53.6 | 61.3 | 54.3 |
| | LayerPruner | 63.9 | 61.6 | 36.7 | 62.3 | 26.6 | 68.4 | 57.4 | 62.4 | 54.9 |
| | Self-AttentionPruner | 69.9 | 76.6 | 47.7 | 50.8 | 30.6 | 77.3 | 58.0 | 61.7 | 59.1 |
| | AggregationPruner | **74.4** | **78.3** | **52.4** | **74.8** | **32.2** | **78.8** | **60.7** | **64.5** | **64.5** |
| 12 | FFNPruner | 72.7 | 62.0 | 31.2 | 63.1 | 21.8 | 63.8 | 57.9 | 63.2 | 54.5 |
| | LayerPruner | 63.6 | 58.2 | 34.0 | 63.3 | 23.6 | 64.6 | 49.0 | 54.7 | 51.4 |
| | Self-AttentionPruner | 70.4 | 75.5 | 45.1 | 49.4 | 29.0 | 75.5 | 57.5 | 61.7 | 58.0 |
| | AggregationPruner | **74.3** | **77.1** | **51.3** | **75.0** | **31.0** | **77.3** | **60.8** | **64.5** | **63.9** |
| 14 | FFNPruner | 71.4 | 62.4 | 29.2 | 62.4 | 18.8 | 61.2 | 59.8 | 62.2 | 53.4 |
| | LayerPruner | 71.8 | 58.7 | 32.1 | 62.2 | 24.4 | 63.1 | 59.6 | 64.2 | 54.5 |
| | Self-AttentionPruner | 67.6 | 75.3 | 44.0 | 49.8 | 27.0 | 75.4 | 57.3 | 60.3 | 57.1 |
| | AggregationPruner | **72.4** | **76.7** | **49.1** | **76.3** | **29.6** | **77.5** | **61.7** | **64.6** | **63.5** |

7B/72B, and Gemma2-9B for TriviaQA. For Mistral-7B-v0.3 on TriviaQA and LLaMA3.1-70B, it shows a slight improvement. Additionally, on GSM8K with LLaMA3.1-70B, Gemma2-9B, and Mistral-7B-v0.3, our performance is comparable to the best baseline. These results consistently demonstrate that our method surpasses the three baselines across various models and tasks. Besides, our results reveal a clear ranking in overall performance among the pruning methods: Aggregation-Pruner > Self-AttentionPruner > FFNPruner > LayerPruner. Notably, FFNPruner and LayerPruner exhibit a rapid decline in performance, dropping to zero as the number of pruned layers increases, compared with the other two methods. These results emphasize the critical importance of transformation parameters in both the FFN and Self-Attention modules for generation tasks. This observation aligns with our claim in Section 3.2.2.

Furthermore, our analysis revealed that as the number of pruned layers increases, the performance of LLMs drops more rapidly on GSM8K compared to TriviaQA. This discrepancy can be attributed to the differing response lengths required for each task. We observed that unpruned LLMs typically encounter the end-of-sequence (EOS) token within 16 tokens when generating answers for TriviaQA. In contrast, GSM8K often requires more (up to 256) tokens to produce a complete answer. Pruned LLMs, which generate one token at a time, are more susceptible to errors than their unpruned counterparts. This vulnerability is exacerbated in tasks requiring longer responses, as each additional token introduces the potential for error accumulation. Consequently, the extended response length needed for GSM8K leads to a more pronounced performance decline in pruned LLMs compared to the shorter responses typical of TriviaQA.

**Discriminative Tasks.** We present the performance of six LLMs on discriminative tasks in Tables 2, 5, 6, and 4. Due to space limit in the main text, Tables 5, 6, and 4 are included in Appendix C. We also report the average performance across eight discriminative tasks. The results demonstrate that our pruning algorithm outperforms the baselines on LLaMa3.1-8B, Qwen2-7B/72B, and Mistral-7B-v0.3, while achieving comparable performance to the best baseline on LLaMA3.1-70B and Gemma2-9B. Notably, as we increase the number of pruned layers, the performance degradation on discriminative tasks is less pronounced compared to generation tasks. This discrepancy can be

attributed to the nature of discriminative tasks, which typically involve multiple-choice questions with limited options, making them inherently simpler than generation tasks that require predicting the next token from the entire vocabulary. To further validate this claim, we conduct additional experiments using a reward model with AggregationPruner. Specifically, we evaluate the Skywork/Skywork-Reward-Llama-3.1-8B model from Hugging Face on RewardBench (Lambert et al., 2024) to assess the impact of pruning algorithm on reward model performance.

Figure 8 in Appendix C illustrates the performance of the reward model on RewardBench. Notably, when 16 layers are pruned, the model's performance remains nearly identical to that of the unpruned version. However, an additional experiment reveals differences when using the pruned and unpruned models to annotate rewards for online alignment (Cen et al., 2024; Dong et al., 2024a). We observe a significant disparity in the reward distributions generated by the pruned and unpruned models. The mean in the rewards gap is 3.53, with a standard deviation of 5.77. This discrepancy can be attributed to the nature of the tasks: RewardBench primarily involves preference choices between two responses, essentially a binary classification problem. In contrast, reward annotation operates on a continuous real number scale, which is a more challenging task. These findings lead us to conclude that pruned models are better suited for maintaining performance on discriminative tasks with limited options. This conclusion makes pruned reward models particularly well-suited for online Direct

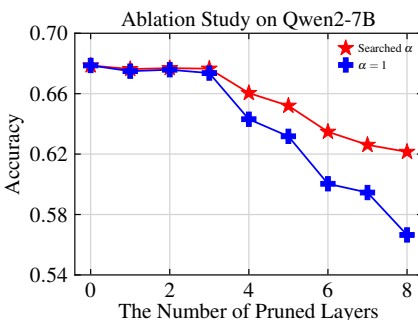

Figure 3: Performance comparison between the default alpha setting ($\alpha = 1$) and the $\alpha$ value obtained through grid search for Qwen2-7B. The experiment evaluates model accuracy averaged across eight discriminative tasks.

Preference Optimization (DPO) (Rafailov et al., 2023) settings. In such settings, each iteration requires only on-policy preference data, and the reduced latency of pruned models is advantageous. However, this same attribute makes them less ideal for online Reinforcement Learning from Human Feedback (RLHF) (Ouyang et al., 2022), where more nuanced reward annotations may be necessary.

### 4.3 ABLATION STUDY

In this section, we evaluate the efficacy of our proposed $\alpha$ search algorithm, as described in Section 3.3. Our experiments focus on Qwen2-7B, and we present the average accuracy across eight discriminative tasks. As illustrated in Figure 3, the alpha value obtained through our grid search method demonstrates better performance compared to the default setting of $\alpha = 1$. These results demonstrate the effectiveness of our algorithm in improving model performance.

## 5 RELATED WORK

**Pruning.** Pruning is a widely adopted and efficient technique in both Computer Vision and Large Language Models. It can be categorized into two main types: Structured Pruning and Unstructured Pruning. Structured Pruning (Lagunas et al., 2021; Xia et al., 2022; Kurtic et al., 2023; He & Xiao, 2023; Xia et al., 2024) involves removing entire filters from neural networks, making it particularly conducive to model deployment. On the other hand, Unstructured Pruning (Chen et al., 2020b; Sanh et al., 2020) focuses on removing individual neurons within the network. Some recent works (Men et al., 2024; Zhong et al., 2024; Gromov et al., 2024; He et al., 2024; Siddiqui et al., 2024) have been proposed to prune blocks in the higher layers of Large Language Models.

## 6 CONCLUSION

In this work, we propose AggregationPruner, a novel approach that focuses on pruning query and key parameters in the higher layers of LLMs. Our method can reduce GPU memory consumption associated with the KV cache during generation tasks. Through extensive experimentation, we demonstrate that our pruning algorithm consistently outperforms recent block pruning techniques, offering a significant advancement in model efficiency without compromising performance. We hope our work will inspire future research on pruning strategies to reduce the KV cache in LLM serving.

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

## A    BENCHMARK DETAILS

For our evaluation on discriminative tasks, we employ the `lm-evaluation-harness` package (version 0.4.2) developed by Gao et al. (2021)[1]. All experiments for discriminative tasks are conducted using an Nvidia A100 80G GPU. It's important to note that the `lm-evaluation-harness` provides two accuracy metrics: "acc" and "acc_norm" for the ARC-Challenge, OpenBookQA, PIQA, and MedQA benchmarks. For these benchmarks, we report the "acc" accuracy results. Table 3 reports the number of tasks and the number of choices for each discriminative task.

In our evaluation of generation tasks, we utilize version 0.4.3 of the `lm-evaluation-harness` package. All experiments for generation tasks are conducted using an Nvidia H100 80G GPU. For the GSM8K and TriviaQA benchmarks, this package offers two accuracy metrics: "exact_match,strict-match" and "exact_match,flexible-extract". In our reporting, we use the "exact_match,strict-match" accuracy results for these benchmarks. The number of tasks for GSM8K and TriviaQA are 1319 and 17944, respectively.

Table 3: Dataset Statistics

| Metric | CommonsenseQA | WinoGrande | ARC-Challenge | BoolQ | OpenBookQA | PIQA | MedQA | MMLU |
|---|---|---|---|---|---|---|---|---|
| # Tasks | 1221 | 1267 | 1172 | 3270 | 500 | 1838 | 1273 | 11973 |
| # Choices | 5 | 2 | 4 | 2 | 4 | 2 | 4 | 4 |

## B    GREEDY SEARCH DETAILS

For the search of optimal $\alpha$, we utilize the wikitext task provided in `lm-evaluation-harness` (version 0.4.2) to compute perplexity. While this package reports three types of perplexity metrics: "word_perplexity", "byte_perplexity", and "bits_per_byte". We employ the "word_perplexity" metric in our search for $\alpha$. The experiments are conducted using one Nvidia A100 80G GPU.

We present the searched alpha values for Mistral-7B-v0.3, Gemma2-9B, LLaMA3.1-8B, Qwen2-7B, LLaMA3.1-70B, and Qwen2-72B in Figures 4, 5, 6, and 7. Our findings reveal that different models yield distinct alpha values for each pruned layer, with layer indices starting at 0.

In Mistral-7B-v0.3, LLaMA3.1-8B, and Qwen2-7B/72B, we observed a trend where the searched alpha values increase as the layer index rises. We hypothesize that this pattern may be attributed to our top-down search approach, resulting in higher alpha values for upper layers.

Conversely, the alpha values searched for Gemma2-9B and LLaMA3.1-70B exhibit fluctuations. The exploration of more sophisticated search methods is left for future research.

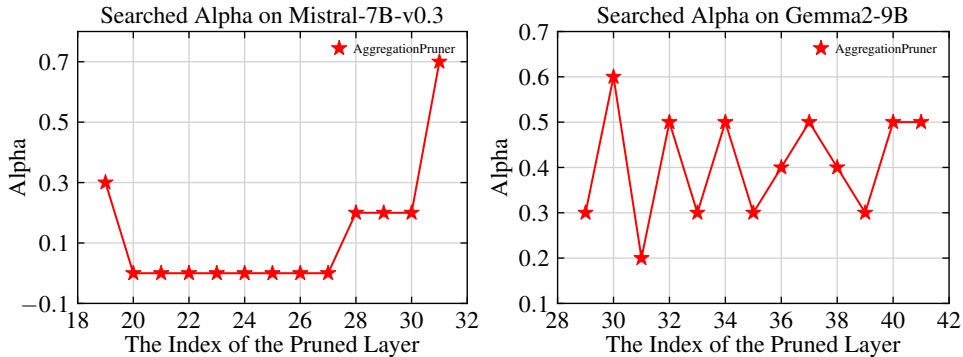

Figure 4: Searched alpha on Mistral-7B-v0.3 and Gemma2-9B.

---

[1]https://github.com/EleutherAI/lm-evaluation-harness

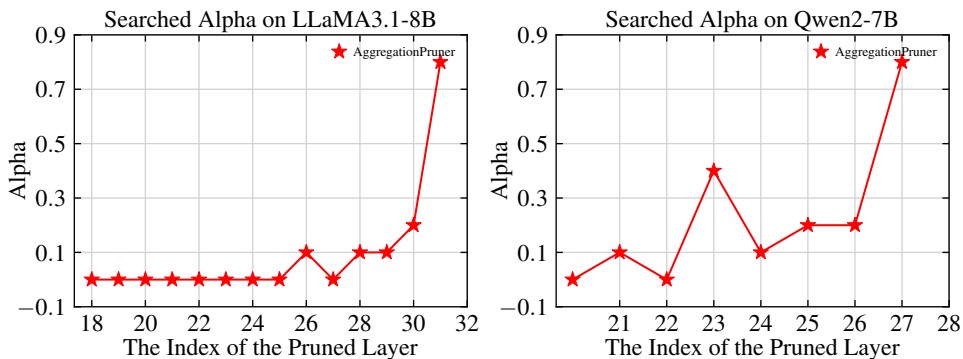

Figure 5: Searched alpha on LLaMA3.1-8B and Qwen2-7B.

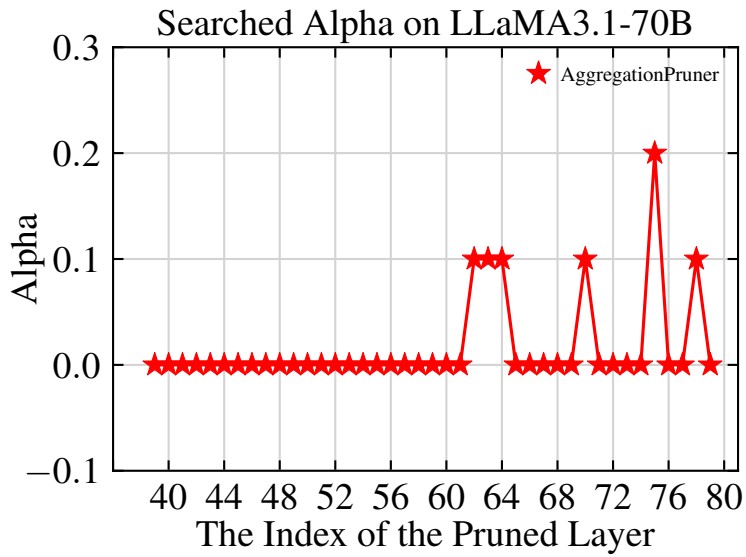

Figure 6: Searched alpha on LLaMA3.1-70B.

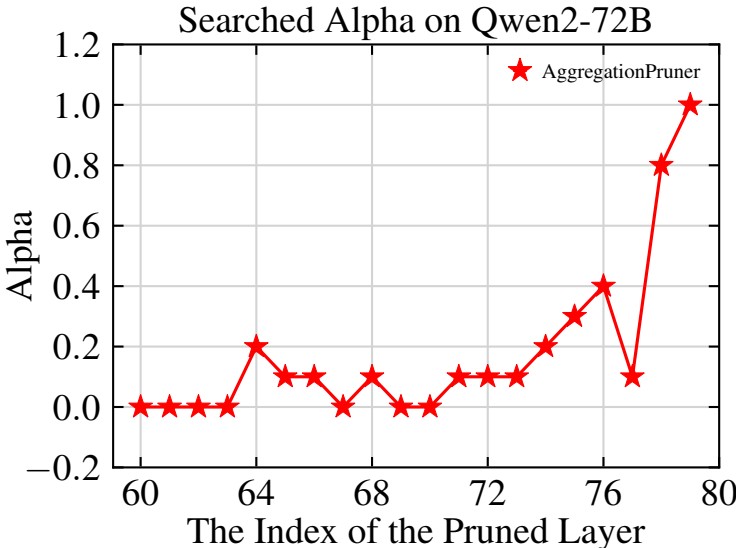

Figure 7: Searched alpha on Qwen2-72B.

# C    MORE EXPERIMENTAL RESULTS

## C.1    DISCRIMINATIVE TASK RESULTS

We present more results on discriminative tasks in Tables 4, 5, and 6.

Table 4: The Performance of Qwen2-7B on Discriminative Tasks.

| | | | | | | | | | | **Qwen2-7B** |
|---|---|---|---|---|---|---|---|---|---|---|
| **#Layers** | **Method** | **CommonsenseQA** | **WinoGrande** | **ARC-Challenge** | **BoolQ** | **OpenBookQA** | **PIQA** | **MedQA** | **MMLU** | **Average** |
| 0 | No Pruning | 80.8 | 77.2 | 58.1 | 84.9 | 34.8 | 79.9 | 56.7 | 70.5 | 67.9 |
| 2 | FFNPruner | **81.3** | 70.1 | 50.2 | 83.9 | 31.6 | 73.8 | 55.5 | **70.0** | 64.6 |
| | LayerPruner | 79.7 | 66.9 | 50.4 | 83.3 | 32.4 | 71.1 | 55.4 | 69.6 | 63.6 |
| | Self-AttentionPruner | 59.6 | 70.6 | 44.6 | 33.4 | 28.2 | 75.4 | 35.3 | 42.9 | 48.8 |
| | AggregationPruner | 80.8 | **76.7** | **57.7** | **84.8** | **34.8** | **80.1** | **56.6** | **70.0** | **67.7** |
| 4 | FFNPruner | **80.5** | 65.5 | 40.1 | 84.5 | 25.8 | 67.7 | 54.6 | 67.3 | 60.8 |
| | LayerPruner | 42.1 | 64.0 | 42.0 | 82.9 | 28.2 | 67.7 | 44.3 | 57.9 | 53.6 |
| | Self-AttentionPruner | 51.6 | 70.5 | 45.1 | 29.5 | 28.4 | 75.4 | 24.2 | 32.3 | 44.6 |
| | AggregationPruner | 71.0 | **76.5** | **57.7** | **84.9** | **35.0** | **79.5** | **55.7** | **68.1** | **66.1** |
| 6 | FFNPruner | **79.0** | 59.4 | 30.1 | 67.8 | 19.8 | 65.0 | 53.7 | 66.8 | 55.2 |
| | LayerPruner | 35.6 | 63.5 | 35.3 | 62.2 | 25.8 | 63.1 | 39.0 | 40.3 | 45.6 |
| | Self-AttentionPruner | 48.6 | 68.8 | 43.8 | 34.0 | 26.2 | 74.5 | 29.7 | 44.0 | 46.2 |
| | AggregationPruner | 57.0 | **76.1** | **54.9** | **84.9** | **33.0** | **78.8** | **56.0** | **67.0** | **63.5** |
| 8 | FFNPruner | 43.8 | 52.6 | 26.5 | 69.7 | 19.6 | 61.8 | 49.8 | 59.3 | 47.9 |
| | LayerPruner | 29.0 | 58.7 | 30.5 | 62.2 | 20.6 | 60.4 | 34.1 | 30.5 | 40.8 |
| | Self-AttentionPruner | 33.7 | 67.0 | 39.1 | 34.1 | 24.2 | 71.8 | 26.1 | 35.4 | 41.4 |
| | AggregationPruner | **58.4** | **74.0** | **51.5** | **85.1** | **29.4** | **75.8** | **56.1** | **66.9** | **62.2** |

## C.2    REWARD MODEL RESULTS

Figure 8 presents our reward model results, which are obtained using RewardBench (Lambert et al., 2024) for evaluation. We conduct these evaluations on an Nvidia H100 80G GPU, utilizing the reward model provided by Skywork[2].

To annotate reward values for our prompt-response data in online alignment setting, we employ a multi-step process. First, we fine-tune the meta-llama/Meta-Llama-3-8B model[3] using an instruction dataset provided by RLHFlow (Dong et al., 2024a)[4]. We then use this instruction-tuned model to generate responses to prompts from the RLHFlow dataset[5], sampling two responses for each prompt. Finally, we annotate these responses with reward values using a reward model that has been pruned by 16 layers using the AggregationPruner method.

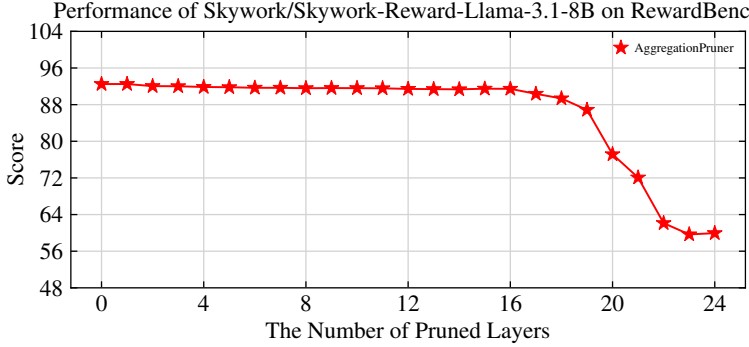

Figure 8: We evaluate the performance of Skywork/Skywork-Reward-Llama-3.1- 8B on Reward-Bench. Our evaluation involves progressively pruning layers, starting from 0 and extending up to 12 layers.

---

[2]https://huggingface.co/Skywork/Skywork-Reward-Llama-3.1-8B

[3]https://huggingface.co/meta-llama/Meta-Llama-3-8B

[4]https://huggingface.co/datasets/RLHFlow/SFT-OpenHermes-2.5-Standard

[5]https://huggingface.co/datasets/RLHFlow/iterative-prompt-v1-iter1-20K

Table 5: The Performance of Mistral-7B-v0.3 and Gemma2-9B on Discriminative Tasks.

| #Layers | Method | CommonsenseQA | WinoGrande | ARC-Challenge | BoolQ | OpenBookQA | PIQA | MedQA | MMLU | Average |
|---|---|---|---|---|---|---|---|---|---|---|
| **Mistral-7B-v0.3** | | | | | | | | | | |
| 0 | No Pruning | 71.7 | 78.5 | 57.7 | 82.2 | 33.6 | 80.5 | 50.1 | 62.2 | 64.6 |
| 2 | FFNPruner | **71.6** | 74.3 | 53.7 | 80.1 | 30.8 | 78.5 | 49.8 | 61.2 | 62.5 |
| | LayerPruner | 71.3 | 72.4 | 52.4 | 81.2 | 31.8 | 78.2 | 48.5 | 60.9 | 62.1 |
| | Self-AttentionPruner | 71.3 | **77.9** | 55.1 | 81.5 | 33.4 | **80.1** | 49.7 | 59.2 | 63.5 |
| | AggregationPruner | **71.6** | **77.9** | **56.8** | **81.6** | **34.4** | 79.9 | **50.0** | **62.1** | **64.3** |
| 4 | FFNPruner | 69.5 | 69.9 | 47.5 | 65.7 | 28.8 | 72.7 | 46.4 | 59.5 | 57.5 |
| | LayerPruner | 69.9 | 68.5 | 47.3 | 65.3 | 28.4 | 69.9 | 44.0 | 57.7 | 56.4 |
| | Self-AttentionPruner | 66.4 | 77.6 | 55.0 | 79.1 | 30.2 | **79.9** | 44.1 | 50.6 | 60.4 |
| | AggregationPruner | **71.7** | **78.0** | **56.3** | **81.8** | **32.8** | 79.9 | **50.0** | **61.5** | **64.0** |
| 6 | FFNPruner | 66.5 | 67.9 | 42.5 | 62.2 | 24.6 | 68.4 | 46.4 | 58.1 | 54.6 |
| | LayerPruner | 69.7 | 65.4 | 40.7 | 62.2 | 25.8 | 66.1 | 47.0 | 61.1 | 54.8 |
| | Self-AttentionPruner | 67.2 | **77.7** | 54.8 | 77.8 | 31.2 | **80.0** | 45.6 | 50.4 | 60.6 |
| | AggregationPruner | **71.7** | 77.6 | **56.1** | **80.7** | **33.2** | 79.8 | 49.0 | **61.6** | **63.7** |
| 8 | FFNPruner | 62.3 | 64.4 | 37.9 | 62.7 | 25.2 | 67.3 | 48.4 | 55.5 | 53.0 |
| | LayerPruner | 68.4 | 62.2 | 37.6 | 63.6 | 24.6 | 64.4 | 48.8 | 57.0 | 53.3 |
| | Self-AttentionPruner | 66.7 | **77.3** | 54.6 | 77.1 | 31.6 | **79.7** | 44.1 | 49.9 | 60.1 |
| | AggregationPruner | **71.4** | **77.3** | **55.8** | **80.6** | **32.8** | 79.1 | 49.1 | **61.4** | **63.4** |
| 10 | FFNPruner | 58.1 | 61.6 | 35.6 | 62.3 | 23.0 | 64.7 | 42.8 | 52.1 | 50.0 |
| | LayerPruner | 64.1 | 60.7 | 35.7 | 62.4 | 23.8 | 63.5 | 48.5 | 56.7 | 51.9 |
| | Self-AttentionPruner | 67.3 | 77.5 | 54.2 | 75.6 | 31.6 | **79.4** | 43.6 | 51.2 | 60.1 |
| | AggregationPruner | **71.4** | **77.6** | **56.6** | **79.7** | **33.4** | 79.0 | 49.3 | **61.5** | **63.6** |
| 12 | FFNPruner | 59.0 | 61.6 | 33.2 | 62.2 | 21.4 | 61.9 | 45.1 | 51.6 | 49.5 |
| | LayerPruner | 68.1 | 65.1 | 34.0 | 62.2 | 23.2 | 61.9 | 48.0 | 57.3 | 52.5 |
| | Self-AttentionPruner | 65.9 | 77.4 | 53.4 | 72.7 | 31.2 | 77.4 | 38.3 | 50.6 | 58.4 |
| | AggregationPruner | **71.5** | **77.8** | **53.9** | **79.0** | **32.0** | **77.8** | 50.9 | **61.7** | **63.1** |
| 13 | FFNPruner | 61.1 | 61.5 | 32.5 | 62.2 | 21.2 | 60.8 | 45.0 | 51.3 | 49.5 |
| | LayerPruner | 38.0 | 63.9 | 33.5 | 62.2 | 21.4 | 60.7 | 29.5 | 40.5 | 43.7 |
| | Self-AttentionPruner | 58.0 | 76.1 | 50.9 | 70.9 | **29.8** | 76.4 | 35.2 | 51.9 | 56.2 |
| | AggregationPruner | **62.9** | **77.0** | **52.6** | **78.2** | 28.8 | **77.5** | 46.6 | **59.8** | **60.4** |
| **Gemma2-9B** | | | | | | | | | | |
| 0 | No Pruning | 77.6 | 80.0 | 64.9 | 84.4 | 33.4 | 81.4 | 60.0 | 70.6 | 69.0 |
| 2 | FFNPruner | 77.1 | 78.1 | 64.6 | 84.0 | 34.6 | 80.8 | 60.4 | 70.5 | 68.8 |
| | LayerPruner | 77.5 | 78.0 | 63.5 | **84.6** | 33.8 | 80.2 | 60.8 | **70.6** | 68.6 |
| | Self-AttentionPruner | 74.9 | **80.0** | 64.5 | 83.7 | **35.6** | **81.5** | 60.9 | 69.1 | 68.8 |
| | AggregationPruner | **77.7** | **80.0** | **65.0** | 83.9 | 34.6 | 81.4 | **61.2** | **70.6** | **69.3** |
| 4 | FFNPruner | 77.0 | 77.2 | 63.0 | **84.3** | 35.4 | 79.5 | 58.8 | 70.4 | 68.2 |
| | LayerPruner | **79.1** | 76.3 | 62.2 | **84.3** | **36.2** | 78.7 | 58.6 | **71.0** | 68.3 |
| | Self-AttentionPruner | 77.1 | 78.9 | 64.3 | 83.7 | 35.4 | 80.7 | 59.7 | 70.5 | 68.8 |
| | AggregationPruner | 77.6 | **81.1** | **65.7** | 83.1 | 35.4 | 80.7 | 60.5 | 70.7 | **69.4** |
| 6 | FFNPruner | 76.2 | 77.0 | 59.1 | 79.8 | 33.4 | 77.3 | 59.9 | 69.6 | 66.5 |
| | LayerPruner | **77.9** | 75.1 | 57.8 | 81.8 | **35.0** | 76.2 | 59.8 | **71.0** | 66.8 |
| | Self-AttentionPruner | 77.0 | **79.4** | 61.9 | 82.6 | 34.2 | 80.3 | **60.5** | 70.4 | 68.3 |
| | AggregationPruner | 77.5 | **79.4** | **64.5** | **82.9** | 34.2 | **80.8** | 60.1 | 70.7 | **68.8** |
| 8 | FFNPruner | 75.3 | 74.8 | 53.2 | 63.1 | 30.8 | 75.5 | 58.3 | **70.2** | 62.6 |
| | LayerPruner | 71.9 | 74.5 | 54.1 | 63.1 | 32.8 | 73.5 | 47.4 | 66.7 | 60.5 |
| | Self-AttentionPruner | 77.0 | 77.8 | 61.7 | **83.1** | 34.8 | 79.9 | 59.8 | 70.1 | **68.0** |
| | AggregationPruner | **77.7** | **79.2** | **63.9** | 70.6 | **35.8** | **80.1** | 60.3 | 70.0 | 67.2 |
| 10 | FFNPruner | 73.7 | 74.3 | 46.2 | 62.7 | 29.4 | 72.3 | 58.4 | 69.3 | 60.8 |
| | LayerPruner | 41.0 | 72.8 | 48.5 | 62.8 | 28.6 | 71.1 | 33.1 | 47.7 | 50.7 |
| | Self-AttentionPruner | 78.0 | 77.8 | 59.8 | **81.2** | 34.2 | 79.6 | **60.6** | 69.7 | **67.6** |
| | AggregationPruner | **78.1** | **78.8** | **62.1** | 55.4 | **35.4** | **79.9** | 59.4 | **69.8** | 64.9 |
| 12 | FFNPruner | 74.4 | 72.4 | 41.7 | 62.6 | 25.4 | 69.4 | 56.6 | 69.1 | 59.0 |
| | LayerPruner | 61.0 | 72.0 | 44.1 | 62.4 | 27.2 | 67.9 | 45.4 | 64.2 | 55.5 |
| | Self-AttentionPruner | 77.4 | 76.7 | 58.5 | **75.7** | 33.4 | 79.2 | 58.8 | 68.7 | **66.0** |
| | AggregationPruner | **78.1** | **78.0** | **61.1** | 57.8 | **34.6** | **79.7** | 59.6 | **70.1** | 64.9 |
| 13 | FFNPruner | 74.3 | 69.9 | 37.5 | 62.8 | 22.2 | 67.7 | 56.5 | 69.4 | 57.5 |
| | LayerPruner | 66.8 | 71.8 | 40.0 | 62.9 | 24.6 | 65.1 | 50.7 | 66.0 | 56.0 |
| | Self-AttentionPruner | 77.1 | 75.8 | 56.3 | **76.4** | 33.4 | 78.7 | 57.3 | 68.2 | **65.4** |
| | AggregationPruner | **77.3** | **77.6** | **59.9** | 59.7 | **35.8** | **79.3** | **60.1** | **70.1** | 65.0 |

Table 6: The Performance of LLaMA3.1-70B and Qwen2-72B on Discriminative Tasks.

| LLaMA3.1-70B | | | | | | | | | |
|---|---|---|---|---|---|---|---|---|---|
| #Layers | Method | CommonsenseQA | WinoGrande | ARC-Challenge | BoolQ | OpenBookQA | PIQA | MedQA | MMLU | Average |
| 0 | No Pruning | 82.6 | 84.1 | 65.4 | 83.9 | 36.6 | 82.3 | 72.7 | 74.4 | 72.8 |
| 8 | FFNPruner | 80.0 | 73.2 | 51.3 | 76.6 | 33.8 | 75.8 | 70.9 | 67.4 | 66.1 |
| | LayerPruner | 81.2 | 72.7 | 50.6 | 79.8 | 34.8 | 73.6 | 66.8 | 70.5 | 66.2 |
| | Self-AttentionPruner | 82.2 | **83.8** | 63.0 | 83.8 | 35.6 | **81.8** | 73.5 | 74.6 | **72.3** |
| | AggregationPruner | **82.5** | 83.7 | **63.1** | 83.8 | 35.6 | 81.7 | **73.6** | 74.6 | **72.3** |
| 16 | FFNPruner | 73.4 | 74.9 | 48.0 | 71.5 | 30.8 | 73.8 | 60.4 | 62.4 | 61.9 |
| | LayerPruner | 80.8 | 68.5 | 49.6 | 75.1 | **34.4** | 72.4 | 72.1 | 73.3 | 65.8 |
| | Self-AttentionPruner | 81.6 | 83.7 | **63.0** | 83.9 | 33.8 | **81.4** | 73.6 | 74.2 | 71.9 |
| | AggregationPruner | **81.7** | **83.9** | 62.7 | 83.9 | **34.4** | 81.0 | **73.8** | **74.3** | 72.0 |
| 24 | FFNPruner | 67.8 | 74.7 | 45.4 | 68.3 | 28.4 | 71.4 | 57.5 | 54.4 | 58.5 |
| | LayerPruner | 54.7 | 69.0 | 47.6 | 73.0 | 33.4 | 71.2 | 50.6 | 63.0 | 57.8 |
| | Self-AttentionPruner | **81.7** | 83.6 | 62.9 | 83.8 | 34.4 | 81.2 | **73.5** | **74.0** | **71.9** |
| | AggregationPruner | 81.4 | **83.8** | **63.0** | **84.1** | **34.6** | **81.3** | 73.4 | **74.0** | **71.9** |
| 32 | FFNPruner | 63.5 | 75.7 | 38.6 | 66.8 | 24.4 | 67.2 | 57.1 | 54.2 | 55.9 |
| | LayerPruner | 70.1 | 66.9 | 42.8 | 72.0 | 30.0 | 68.2 | 65.8 | 69.7 | 60.7 |
| | Self-AttentionPruner | 80.4 | 83.5 | **62.5** | 83.6 | **36.6** | **81.5** | 73.4 | 73.7 | **71.9** |
| | AggregationPruner | **80.6** | **83.7** | 62.3 | **83.8** | 36.4 | 81.2 | **73.5** | **74.0** | **71.9** |
| 40 | FFNPruner | 60.0 | 73.2 | 31.3 | 62.1 | 20.6 | 63.2 | 60.9 | 51.5 | 52.8 |
| | LayerPruner | 63.5 | 70.4 | 38.1 | 62.0 | 24.6 | 65.9 | 53.7 | 65.2 | 55.4 |
| | Self-AttentionPruner | **80.9** | 82.0 | 61.5 | **84.0** | **34.4** | 80.3 | 72.9 | 73.7 | 71.2 |
| | AggregationPruner | 80.3 | **82.7** | **61.9** | **84.0** | 34.2 | **80.4** | **73.5** | **74.2** | **71.4** |
| 41 | FFNPruner | 55.2 | 71.8 | 30.6 | 62.1 | 20.6 | 63.0 | 60.3 | 49.0 | 51.6 |
| | LayerPruner | 40.4 | 70.3 | 37.8 | 62.0 | 23.0 | 65.3 | 55.8 | 60.7 | 51.9 |
| | Self-AttentionPruner | **71.3** | 82.7 | 59.0 | **84.2** | 34.8 | **79.7** | **72.0** | 72.7 | **69.6** |
| | AggregationPruner | 70.9 | **82.8** | **59.7** | 84.0 | 33.4 | **79.7** | **72.0** | **73.1** | 69.4 |
| Qwen2-72B | | | | | | | | | |
| #Layers | Method | CommonsenseQA | WinoGrande | ARC-Challenge | BoolQ | OpenBookQA | PIQA | MedQA | MMLU | Average |
| 0 | No Pruning | 89.3 | 84.5 | 65.3 | 88.3 | 36.2 | 82.4 | 75.6 | 83.8 | 75.7 |
| 3 | FFNPruner | **89.4** | 75.6 | 58.6 | 83.4 | 35.0 | 78.8 | 75.7 | **83.3** | 72.5 |
| | LayerPruner | 88.6 | 72.4 | 57.7 | 84.6 | **36.8** | 77.2 | 74.9 | 82.6 | 71.8 |
| | Self-AttentionPruner | 88.6 | **84.8** | 63.9 | 81.0 | 35.4 | 81.6 | **76.7** | 82.2 | 74.3 |
| | AggregationPruner | 88.9 | 84.2 | **64.4** | **88.5** | 36.4 | **82.3** | 76.5 | 82.8 | **75.5** |
| 6 | FFNPruner | 88.3 | 70.7 | 53.2 | 68.9 | 32.4 | 75.6 | 76.0 | 82.5 | 68.4 |
| | LayerPruner | 88.5 | 69.6 | 53.4 | 76.8 | 31.6 | 73.0 | 74.0 | 81.6 | 68.6 |
| | Self-AttentionPruner | 88.1 | **84.3** | 63.0 | 79.9 | 34.6 | 80.6 | 76.2 | 81.5 | 73.5 |
| | AggregationPruner | **88.7** | 83.9 | **63.7** | **88.3** | **36.4** | **81.7** | **76.7** | **82.7** | **75.3** |
| 9 | FFNPruner | 87.9 | 69.1 | 43.9 | 62.2 | 30.0 | 72.1 | 75.3 | 82.1 | 65.3 |
| | LayerPruner | 86.8 | 67.9 | 48.5 | 62.3 | 31.6 | 70.3 | 73.4 | 80.5 | 65.2 |
| | Self-AttentionPruner | 87.8 | **84.3** | 62.8 | 79.8 | 34.0 | 80.4 | 74.5 | 81.3 | 73.1 |
| | AggregationPruner | **88.9** | 84.2 | **64.1** | **87.8** | **37.0** | **81.7** | **76.4** | **82.8** | **75.4** |
| 12 | FFNPruner | 87.5 | 65.4 | 36.7 | 62.5 | 25.8 | 69.7 | 74.7 | 81.8 | 63.0 |
| | LayerPruner | 84.2 | 67.9 | 45.6 | 64.6 | 29.0 | 69.5 | 75.1 | 81.9 | 64.7 |
| | Self-AttentionPruner | 87.3 | 83.5 | 62.7 | 81.6 | 33.6 | 79.7 | 74.5 | 81.2 | 73.0 |
| | AggregationPruner | **88.9** | **84.1** | **63.7** | **88.2** | **36.6** | **81.0** | **76.4** | **82.9** | **75.2** |
| 15 | FFNPruner | **86.9** | 63.2 | 32.9 | 62.4 | 22.2 | 65.8 | 74.6 | 81.9 | 61.2 |
| | LayerPruner | 74.0 | 68.8 | 43.4 | 64.4 | 26.4 | 67.6 | 74.1 | 79.8 | 62.3 |
| | Self-AttentionPruner | 88.2 | 82.8 | 61.5 | 78.0 | 31.4 | 80.3 | 74.4 | 81.6 | 72.4 |
| | AggregationPruner | 88.4 | **83.4** | **64.2** | **86.7** | **36.2** | **80.6** | **76.4** | **82.9** | **74.9** |
| 18 | FFNPruner | 86.3 | 61.6 | 29.2 | 62.2 | 20.4 | 64.7 | 71.6 | 81.1 | 59.6 |
| | LayerPruner | 76.3 | 68.8 | 39.5 | 64.0 | 24.2 | 64.5 | 74.2 | 79.2 | 61.3 |
| | Self-AttentionPruner | **88.0** | 81.3 | 59.3 | 74.6 | 29.8 | 78.6 | 75.3 | 81.1 | 71.0 |
| | AggregationPruner | **88.0** | **81.8** | **62.5** | **87.7** | **34.8** | **79.9** | **77.1** | **82.5** | **74.3** |
| 19 | FFNPruner | 87.0 | 61.7 | 30.4 | 62.2 | 20.4 | 64.9 | 72.4 | 81.1 | 60.0 |
| | LayerPruner | 72.6 | 68.3 | 38.3 | 63.7 | 23.2 | 64.0 | 72.4 | 78.4 | 60.1 |
| | Self-AttentionPruner | 87.6 | 81.2 | 59.0 | 76.5 | 28.2 | 78.0 | 74.9 | 80.8 | 70.8 |
| | AggregationPruner | **88.1** | **81.8** | **61.7** | **88.1** | **34.8** | **80.0** | **76.4** | **82.3** | **74.2** |

# D    DEMONSTRATION EXAMPLES ON GENERATION TASKS

In this section, we provide some demonstration examples on generation tasks with various pruning algorithms.

Table 7: We present a demonstration example of outputs from various pruning algorithms applied to the LLaMA3.1-70B model. In this demonstration, we prune the last two layers of the model using different pruning methods. The comparative results are shown using the TriviaQA task. We can find that LayerPruner produces incorrect answers, while AggregationPruner and Self-AttentionPruner provide the correct ones.

```
Prompt:  Which feminist book label was established by Carmen Callil
and others in 1973?
Answer:  Viragos
Question:  What is the name of the thoroughfare that Harry Potter
lived with his Uncle's family?
Answer:  Eeylops Owl Emporium
Question:  Plaid Cymru (roughly pronounced 'plied cumrie') is the
nationalist political party of which nation?
Answer:  Welsh nation
Question:  Thomas Becket was murdered where?
Answer:  Our Lady of the Undercroft
Question:  How many countries make up Europe?
Answer:  forty-eight
Question:  What claimed the life of singer Kathleen Ferrier?
Answer:
```

**AggregationPruner:**  Cancer

**Self-AttentionPruner:**  Cancer

**LayerPruner:**  Ovarian cancer claimed her life at age41 in1953Ž0331953

**FFNPruner:**  cancerous growths in her voicebox

Table 8: We present a demonstration example of outputs from various pruning algorithms applied to the Qwen2-72B model. In this demonstration, we prune the last 12 layers of the model for AggregationPruner and Self-AttentionPruner. We prune the last 5 and 7 layers of the model for LayerPruner and FFNPruner The comparative results are shown using the GSM8K task. We find that FFNPruner and LayerPruner produce redundant content and fail to generate the correct answer. On the other hand, AggregationPruner and Self-AttentionPruner both arrive at the correct solution, albeit with differing outputs.

**Prompt:** Question: Steve finds 100 gold bars while visiting Oregon. He wants to distribute his gold bars evenly to his 4 friends. If 20 gold bars were lost on the way back to San Diego, how many gold bars will each of his 4 friends get when he returns?
Answer: He only has $100 - 20 = 80$ gold bars after losing 20 of them.
He then gives each of his friends $\frac{80}{4} = 20$ gold bars.
#### 20
Question: In a week, Mortdecai collects 8 dozen eggs every Tuesday and Thursday, and he delivers 3 dozen eggs to the market and 5 dozen eggs to the mall. He then uses 4 dozen eggs to make a pie every Saturday. Mortdecai donates the remaining eggs to the charity by Sunday. How many eggs does he donate to the charity?
Answer: Mortdecai collects a total of $8 \times 2 = 16$ dozens of eggs.
He sells a total of $3 + 5 = 8$ dozens of eggs.
So, $16 - 8 = 8$ dozens of eggs are left.
After using 4 dozens of eggs to make a pie, $8 - 4 = 4$ dozens of eggs are left.
Since there are 12 in 1 dozen, Mortdecai donates $4 \times 12 = 48$ pieces of eggs to the charity.
#### 48
Question: Corey downloaded two movie series from his Netflix account with 12 and 14 seasons per series, respectively. However, in the week, his computer got a mechanical failure, and he lost two episodes from each season for both series. If each season in the movie series that Corey downloaded had 16 episodes, how many episodes remained after the computer's mechanical failure?
Answer: In the first movie series with 12 seasons, after the mechanical failure, the number of episodes that Corey lost is $2 \times 1224$.
Originally, the movie series with 12 seasons had $12 \times 16 = 192$ episodes.
After the mechanical failure, Corey had $192 - 24 = 168$ episodes remaining in the first movie series.
Similarly, the 14-season movie series also had $14 \times 2 = 28$ episodes lost after the computer's mechanical failure.
Originally, the movie series with 14 seasons had $14 \times 16 = 224$ episodes.
The mechanical failure of the computer reduced the number of episodes in the 14-season movie series to $224 - 28 = 196$.
After the loss, Corey had $196 + 168 = 364$ episodes remaining from the two movie series he had downloaded.
#### 364
Question: There were 18 students assigned in a minibus for a field trip. Eight of these students were boys. On the day of the field trip, the number of girls and boys was the same since some of the girls were not able to join the trip. How many girls were not able to join the field trip?
Answer: $8$ boys $+ 8$ girls $= 16$ students joined the field trip.
Thus, $18 - 16 = 2$ girls were not able to join the field trip.
#### 2
Question: There are 200 more red apples than green apples in a grocery store. A truck arrives and delivers another 340 green apples. If there were originally 32 green apples, how many more green apples than red apples are there in the store now?
Answer: There are $200 + 32 = 232$ red apples.
After the delivery, there are $340 + 32 = 372$ green apples.
There are now $372 - 232 = 140$ more green apples than red apples.
#### 140
Question: A robe takes 2 bolts of blue fiber and half that much white fiber. How many bolts in total does it take?
Answer:
**AggregationPruner:** The white fibers take $\frac{2}{2} = 1$ pound per bookend.
So, in total, it takes $2 + 1 = 3$ pounds of fiber per robe.
#### 3

**Self-AttentionPruner:** The number of white fibers is calculated using:

$$\text{COUNTIFS("A2:A10", ">0")}$$

The total number of bolts is calculated using:

$$\text{=COUNTIFS("A2:A10", ">0")}$$

#### 3

**LayerPruner:** white fiber plus two blue fiber equals total fiber used 2+ half of two equals total fiber used 2+ half of two equals total fiber used 2+ half of two equals total fiber used two plus half of two equals total fiber used two plus half of two equals total fiber used two plus half of two equals total fiber used two plus half of two equals total fiber used two plus half of two equals total fiber used two plus half of two equals total fiber used two plus half of two equals total fiber used two plus half of two equals total fiber used two plus half of two equals total fiber used

**FFNPruner:** White fiber + 2 bolts of blue fiber = $\langle\langle$ 2 + 2 = 4 $\langle\langle$ 4 bolts total fiber.
White fiber = $\frac{2}{2} = \langle\langle 2/2 = 1\rangle\rangle$ 1 bolt of white fiber.
White fiber + blue fiber = $\langle\langle 1 + 2 = 3\rangle\rangle$3 bolts total fiber.
White fiber + blue fiber = $\langle\langle 1 + 2 = 3\rangle\rangle$3 bolts total fiber.
White fiber + blue fiber = $\langle\langle 1 + 2 = 3\rangle\rangle$3 bolts total fiber.
White fiber + blue fiber = $\langle\langle 1 + 2 = 3\rangle\rangle$3 bolts total fiber.
White fiber + blue fiber = $\langle\langle 1 + 2 = 3\rangle\rangle$3 bolts total fiber.
White fiber + blue fiber = $\langle\langle 1 + 2 = 3\rangle\rangle$3 bolts total fiber.
White fiber + blue fiber = $\langle\langle 1 + 2 = 3\rangle\rangle$3 bolts total fiber.

