# OpenReview forum: "Pruning Aggregation Parameters for Large Language Models"
_ICLR.cc/2025/Conference — Submitted to ICLR 2025_

### Official Review · Reviewer_o9F7 · 2024-10-31

**Soundness:** 3
**Presentation:** 3
**Contribution:** 2
**Rating:** 6
**Confidence:** 4

**Summary:**

The study introduces a novel pruning algorithm that specifically targets aggregation parameters within large language models to reduce the model size and lower GPU memory usage during inference. By incorporating a rescaling parameter,  the method enhances the performance of pruned models without additional training.

**Strengths:**

1. This work proposes a novel pruning algorithm that requires no additional training and targets specific parameters within LLMs.
2. This work introduces a rescaling parameter that adjusts the output of the pruned block to further improve the performance of the pruned LLM.
3. Extensive experiments demonstrate that the proposed method outperforms the recent block pruning algorithms.

**Weaknesses:**

1. While the experimental results support the effectiveness of the proposed method, the paper lacks theoretical analysis on why pruning aggregation parameters has minimal impact on model performance. The authors should provide more theoretical support or in-depth analysis.
2. The paper bases the selection of the rescaling parameter α on grid search but does not discuss its impact on model performance in detail. The authors should further explore how the choice of α affects model performance.
3. The paper focuses on model compression and acceleration but few discuss the generalization of pruned models across different domain tasks.
4. The paper lessly discusses the model's performance and efficiency on actual hardware.
5. To promote the study's reproducibility, authors are recommended to provide the code used in the experiment and the preprocessed data so that other researchers can reproduce the results.

**Questions:**

1. Existing methods bring additional training and huge computational overhead. What is the extra cost? What is the cost comparison between the proposed method and the existing method?
2. Why do existing methods maintain performance in domains not well covered by additional training data? What was the experiment? Analyze the specific reasons.
3. What is the key problem to be solved in this paper?

---

> ### Author Response · Authors · 2024-11-16
>
> **Q1**: While the experimental results support the effectiveness of the proposed method, the paper lacks theoretical analysis on why pruning aggregation parameters has minimal impact on model performance. The authors should provide more theoretical support or in-depth analysis.
>
> **A1**:  Previous studies [1,2] have theoretically demonstrated the issue of over-smoothing in Transformers. These works suggest that removing aggregation layers can help mitigate over-smoothing, indicating that excessive aggregation may be detrimental. Our study builds upon these theoretical foundations, though we focus primarily on other contributions; hence, we omit the detailed theoretical analysis presented in previous works. [3,4] have experimentally demonstrated the roles of various parameter types, and our work builds upon their findings.
>
> **Q2**: The paper bases the selection of the rescaling parameter α on grid search but does not discuss its impact on model performance in detail. The authors should further explore how the choice of α affects model performance.
>
> **A2**: Figure 3 demonstrates the impact of different alpha values on the results. The alpha value we find through our search outperforms the default alpha of 1 commonly used in LLMs.
>
> **Q3**: The paper focuses on model compression and acceleration but few discuss the generalization of pruned models across different domain tasks.
>
> **A3**: We conduct experiments across 10 benchmarks spanning various domains, where our method consistently outperforms the baselines.
>
> **Q4**: The paper lessly discusses the model's performance and efficiency on actual hardware.
>
> **A4**: Using the LLaMA3.1-8B model, we compare the GPU memory consumption (on Nvidia A100 80G) of AggregationPruner and baseline methods on the GSM8K task when pruning six layers. Our results show that AggregationPruner, LayerPruner, and Self-AttentionPruner consume 17,277 MB, while FFNPruner consumes 45,277 MB. This difference arises because, during implementation, we load all the LLM weights onto the GPU and modify the computation process to achieve pruning. Consequently, AggregationPruner, LayerPruner, and Self-AttentionPruner effectively reduce the KV cache, whereas FFNPruner could not, leading to significantly higher memory consumption. Nonetheless, as illustrated in the second subfigure of Figure 2, our method significantly outperforms the baselines when pruning the same number of layers.
>
> **Q5**: To promote the study's reproducibility, authors are recommended to provide the code used in the experiment and the preprocessed data so that other researchers can reproduce the results.
>
> **A5**: We will release the codes if the paper is accepted.
>
> **Q6**: Existing methods bring additional training and huge computational overhead. What is the extra cost? What is the cost comparison between the proposed method and the existing method?
>
> **A6**: Sheared LLaMA [5] requires additional training on 50B tokens, whereas our method is entirely training-free, incurring no additional training cost.
>
>
> **Q7**: Why do existing methods maintain performance in domains not well covered by additional training data? What was the experiment? Analyze the specific reasons.
>
> **A7**: Since our method is training-free, there is no need for additional training to identify the optimal pruning scheme. This ensures that the pruning process is unaffected by the domain coverage of the training dataset. We conduct experiments across 10 benchmarks spanning various domains, where our method consistently outperforms the baselines.
>
> **Q8**: What is the key problem to be solved in this paper?
>
> **A8**: We propose a pruning algorithm for KV cache reduction.
>
> [1] Revisiting Over-Smoothing in BERT from the Perspective of Graphs
>
> [2] Mitigating Over-Smoothing in Transformers via Regularized Nonlocal Functionals
>
> [3] Transformer feed-forward layers are key-value memories
>
> [4] Mass-editing memory in a transformer
>
> [5] Sheared LLaMA: Accelerating Language Model Pre-training via Structured Pruning

---

> > ### Comment · Reviewer_o9F7 · 2024-11-25
> >
> > I carefully read the comments of the other reviewers and the rebuttal of the authors. The authors gave detailed rebuttals to my questions. I think all my concerns have been well addressed. I choose to keep my rate unchanged.

---

> > > ### Author Response · Authors · 2024-11-25
> > >
> > > Thank you for your suggestions. We would be happy to address any further questions you may have.

---

### Official Review · Reviewer_UEim · 2024-11-03

**Soundness:** 3
**Presentation:** 3
**Contribution:** 3
**Rating:** 6
**Confidence:** 4

**Summary:**

The paper introduces a pruning algorithm (AggregationPruner) which prunes the Key and Value weights (aggregation parameters) of higher layers of LLMs to reduce the GPU memory consumption during LLM inference tasks (primarily during generative tasks). The method (being post training-free) outperforms the baselines in retaining the performance on various benchmark datasets and models.

**Strengths:**

1. The paper is clearly motivated, very well structured and has a great value for LLM inference community.
2. The connection between GNNs and LLMs and various design choices (why they picked only higher layers, picking aggregation matrices) are well established.
3. The analysis that pruned models might be good for discriminative tasks while have effect on generative tasks (Fig 8) is pretty interesting and sort of intuitive. (Infact, I believe this statement holds true for any compression method given the nature/complexity of these tasks but might need validation). That being said, I've a question wrto AggregationPruner, refer 1st point in weakness

**Weaknesses:**

These are not exactly weaknesses per se, but some thoughts I've to improve the paper.

1. [**Experiment**] Can the authors have an experiment to truly validate the claim that picking lower layers will degrade the performance due to the importance of the layers (line 3 in Algorithm 4)? While the math is quite intuitive from GNNs, an experiment/ablation on all layers (0-32) for model on various tasks can be a good addition?
2. [**Comment**] Continuing from previous point, can the authors comment on design choice on choosing the number of layers for real time usage of the method (line 3 in Algorithm 4)? From Fig 2, it seems to be having a lot of variation for different models/datasets.
3. [**Comment**] As mentioned in L419-420, KV cache doesn't get created for discriminative tasks. So, can the authors explain how effective AggregationPruner is for discriminative tasks? I believe authors might have to consider explaining this part more clearly in the paper!
4. [**Experiment**] The paper has been motivated to be effective on GPU memory consumption on generative tasks. Can the authors provide an actual experiment to demonstrate this?
    - *For eg*: Pick a model, a generative task, measure the perplexity/wall-clock time/speed/memory for the baseline model vs AggregratedPruner method. If an experiment of this sort can't be perform, please comment on why that might be the case.
5. [**Experiment/Comment**] L144-146: The authors have mentioned that "given the black-box nature, we choose the KV matrices" and demonstrated the results in the paper. And I also believe Self-AttentionPruner, FFNPruner and LayerPruner are the terms introduced in this paper (please correct me if I am wrong and cite these methods accordingly).
   - Now, given these key design choices, I would like to know if it's possible to report the sparsity ratio for these various methods? For eg: AggregationPruner might be better compared to FFNPruner because the number of parameters being pruned is very-very less in the former (similar to [1]). While I am not expecting an apples-apples comparison in terms of pruned parameters, these details will be informative.
    - I am also aware that the total number of parameters to be pruned in AggregationPruner might be different depending on the size of KV cache which will vary; so the authors can make assumptions while reporting results in the tables. If it's not possible for some reason, please comment.
6. [**Experiment**] Can the authors perform an experiment on the choice of dataset on Greedy Search of Alpha? i.e how dependent the alpha values are with respect to dataset. Suppose it turns out to be dataset dependent, then is it safe to say that the method is calibration-dataset dependent? If so, I believe addition of these details (either in Appendix or main section) will be beneficial.

**Questions:**

**Possible Relevant citation**
- L464 - L468: Following the point 5 from weakness section, while not the exact relation is established, I believe this study [1] has some connection with respect to compressing the FFN blocks vs Attention blocks and the number of parameters involved while pruning different blocks.

[1] The Cost of Compression: Investigating the Impact of Compression on Parametric Knowledge in Language Models - https://arxiv.org/abs/2312.00960

**Format**
1. [**Typo**] - L366: It is FFNPruner
2. [**Presentation Suggestion**] The authors might consider a small block diagram explaining the difference between FFNPruner, LayerPruner, AttentionPruner and AggregationPruner. Or maybe a diagram for the algorithm
3. [**Presentation Suggestion**] It seems like the assumptions/claims on GPU memory consumption has been mentioned at different places without an experiment to validate. So, the authors might reconsider text formatting.

**Note:** Suggestions are not mandatory improvements and the authors can wish to ignore it totally.

---

> ### Author Response · Authors · 2024-11-16
>
> **Q1**. Can the authors have an experiment to truly validate the claim that picking lower layers will degrade the performance due to the importance of the layers
>
> **A1**. We use the AggregationPruner to conduct our experiments, setting all alpha values to 0.1. Specifically, we prune the last 10 layers and the first 10 layers of the Mistral-7B-v0.3 model.
>
> | **Model (bfloat16)**    | **Pruning Strategy**   | **Commonsense QA** | **Winogrande** | **ARC Challenge** | **BoolQ** | **OpenBookQA** | **PIQA** | **MedQA (4 options)** | **MMLU** |
> |--------------------------|-------------------------|--------------------|----------------|-------------------|-----------|----------------|----------|-----------------------|---------|
> | **Mistral-7B-v0.3**    | Drop last 10 layers     | 0.7052             | 0.7553         | 0.4659            | 0.8076    | 0.2700         | 0.7203   | 0.4682                | 0.6142  |
> | **Mistral-7B-v0.3**     | Drop first 10 layers    | 0.1957             | 0.5154         | 0.2116            | 0.3783    | 0.1200         | 0.5419   | 0.2773                | 0.2295  |
>
>
> The experiments reveal that pruning the lower layers significantly degrades performance.
>
> **Q2**: Continuing from previous point, can the authors comment on design choice on choosing the number of layers for real time usage of the method (line 3 in Algorithm 4)? From Fig 2, it seems to be having a lot of variation for different models/datasets.
>
> **A2**: The number of layers that can be pruned depends on the model's tolerance and the nature of the task. For tasks demanding high precision, such as generating exact numerical reward values for responses, pruning too many layers is not feasible. However, for simpler tasks like binary classification, a larger number of layers can be pruned. Additionally, the optimal number of pruned layers varies by model. Conducting preliminary experiments can help establish a reasonable range, which can then be fine-tuned for real-time systems.
>
> **Q3**: As mentioned in L419-420, KV cache doesn't get created for discriminative tasks. So, can the authors explain how effective AggregationPruner is for discriminative tasks? I believe authors might have to consider explaining this part more clearly in the paper!
>
> **A3**: The results of the AggregationPruner on the discriminative task are presented in Tables 2, 4, 5, and 6. Our AggregationPruner demonstrates superior performance compared to the baseline methods. When pruning the same number of layers, our method outperforms the baselines by a significant margin.
>
> **Q4**: The paper has been motivated to be effective on GPU memory consumption on generative tasks. Can the authors provide an actual experiment to demonstrate this?
>
> **A4**: Using the LLaMA3.1-8B model, we compare the GPU memory consumption of AggregationPruner and baseline methods on the GSM8K task when pruning six layers. Our results show that AggregationPruner, LayerPruner, and Self-AttentionPruner consume 17,277 MB, while FFNPruner consumes 45,277 MB. This difference arises because, during implementation, we load all the LLM weights onto the GPU and modify the computation process to achieve pruning. Consequently, AggregationPruner, LayerPruner, and Self-AttentionPruner effectively reduce the KV cache, whereas FFNPruner could not, leading to significantly higher memory consumption. Nonetheless, as illustrated in the second subfigure of Figure 2, our method significantly outperforms the baselines when pruning the same number of layers.
>
> **Q5**: The authors have mentioned that "given the black-box nature, we choose the KV matrices" and demonstrated the results in the paper. And I also believe Self-AttentionPruner, FFNPruner and LayerPruner are the terms introduced in this paper (please correct me if I am wrong and cite these methods accordingly).
>
>
> **A5**: Yes, we summarize the methods in these papers and introduce these terms.

---

> > ### Author Response · Authors · 2024-11-16
> >
> > **Q6**: Now, given these key design choices, I would like to know if it's possible to report the sparsity ratio for these various methods?
> >
> > **A6**: We provide the parameter count in LLaMA3.1-8B for each linear in each layer:
> >
> > | Layer | Parameter Count |
> > | :--- | :--- |
> > | gate_proj | $58,720,256$ |
> > | up_proj | $58,720,256$ |
> > | down_proj | $58,720,256$ |
> > | q_proj | $16,777,216$ |
> > | k_proj | $4,194,304$ |
> > | v_proj | $4,194,304$ |
> > | O_proj | $16,777,216$ |
> >
> > For AggregationPruner, we can achieve the sparsity ratio in one layer:
> > \begin{gathered}
> > \text { Proportion of } Q+K=\frac{Q+K}{\text { Total Layer Parameters }} \times 100, \\
> > \text { Proportion }=\frac{20,971,520}{218,103,232} \times 100 \approx 9.6/100
> > \end{gathered}
> >
> > For Self-AttentionPruner, we can achieve the sparsity ratio in one layer:
> > \begin{aligned}
> > &\text { Proportion of } \mathrm{Q}+\mathrm{K}+\mathrm{V}+\mathrm{O}=\frac{Q+K+V+O}{\text { Total Layer Parameters }} \times 100,\\
> > &\text { Proportion }=\frac{41,943,040}{218,103,232} \times 100 \approx 19.2/100
> > \end{aligned}
> >
> > For FFNPruner, we can achieve the sparsity ratio in one layer:
> > \begin{aligned}
> > &\text { Proportion of } \mathrm{gate_{proj}}+\mathrm{up_{proj}}+\mathrm{down_{proj}}+=\frac{gate_{proj}+up_{proj}+down_{proj}}{\text { Total Layer Parameters }} \times 100,\\
> > &\text { Proportion }=\frac{176,160,768}{218,103,232} \times 100 \approx 80.8/100
> > \end{aligned}
> >
> > For LayerPruner, we can achieve the sparsity ratio in one layer:
> > \begin{aligned}
> > 100/100
> > \end{aligned}
> >
> > **Q7**: I am also aware that the total number of parameters to be pruned in AggregationPruner might be different depending on the size of KV cache which will vary; so the authors can make assumptions while reporting results in the tables. If it's not possible for some reason, please comment.
> >
> > **A7**: Yes, the reduction in KV cache usage depends on the number of layers pruned, as the KV cache is maintained in every layer. Therefore, if we want to reduce more kv cache, we need to prune more layers.
> >
> > **Q8**: Can the authors perform an experiment on the choice of dataset on Greedy Search of Alpha? i.e how dependent the alpha values are with respect to dataset. Suppose it turns out to be dataset dependent, then is it safe to say that the method is calibration-dataset dependent? If so, I believe addition of these details (either in Appendix or main section) will be beneficial.
> >
> > **A8**: We use the pile_10k dataset (https://github.com/EleutherAI/lm-evaluation-harness/blob/badf273aa08c6d5b2d0b92e247f7b50aaa4fb6da/lm_eval/tasks/pile/pile_arxiv.yaml#L11) from the lm-evaluation-harness package to search for the alpha value in LLaMA3.1-8B. Our results show that we get the same alpha value (compared with wikitext dataset) for the pruned layers.
> >
> >
> > **Q9**: Citation and Presentation.
> >
> > **A10**: We will polish the paper based on the suggestion in the next version.

---

> > > ### Comment · Reviewer_UEim · 2024-11-21
> > > **Review for rebuttal**
> > >
> > > Thanks for the detailed rebuttal. The authors have answered my questions and I would like to maintain my score.
> > >
> > > Polishing the paper based on this (and others feedback) will strengthen this paper. All the best with your submission.

---

> > > > ### Author Response · Authors · 2024-11-21
> > > >
> > > > Thank you for your very helpful suggestions and kind words. We would be happy to address any further questions you may have. Thanks!

---

### Official Review · Reviewer_Kwoc · 2024-11-04

**Soundness:** 2
**Presentation:** 2
**Contribution:** 2
**Rating:** 3
**Confidence:** 4

**Summary:**

This paper proposes a methodology for pruning LLMs without needing to do any fine-tuning of the model, by targeting "aggregation" parameters, as defined by the authors, in the later layers of the mode. By doing so, the KV-cache can be reduced in size, which should result in cheaper inference. The authors demonstrate that accuracy losses for several popular LLMs can be manageable with this methodology.

**Strengths:**

Pruning techniques that don't require fine-tuning the model are always welcome, and this technique does seem to be able to change many of the layers in popular LLMs without hurting the accuracy too much. I also thought the reasoning as to why the authors wanted to prune these particular parameters was interesting.

**Weaknesses:**

The biggest problem with this paper is that it presents no quantitative data as to whether inference using these pruned models is actually cheaper or faster than traditional inference, and it doesn't contextualize the results against other ways of making a model cheaper to run. The strongest claim in the paper is that the KV-cache size can be reduced, which is useful, but that is not the same as a speedup. The authors did not compare against pruning methodologies that require fine-tuning, or against other forms of quantization or sparsity. Techniques like these have to be understood in context, and the paper doesn't provide it.

**Questions:**

1. What is the measured speedup of this approach, especially when changing the batch size to take advantage of the KV-cache compression.
2. Did you evaluate using BF16 number formats? Could you compare against PTQ with FP8?
3. Could you compare against other pruning approaches?

---

> ### Author Response · Authors · 2024-11-12
>
> W1: The biggest problem with this paper is that it presents no quantitative data as to whether inference using these pruned models is actually cheaper or faster than traditional inference, and it doesn't contextualize the results against other ways of making a model cheaper to run. The strongest claim in the paper is that the KV-cache size can be reduced, which is useful, but that is not the same as a speedup. The authors did not compare against pruning methodologies that require fine-tuning, or against other forms of quantization or sparsity. Techniques like these have to be understood in context, and the paper doesn't provide it.  Q1: What is the measured speedup of this approach, especially when changing the batch size to take advantage of the KV-cache compression.
>
> A1: Please note that we focus on enhancing throughput in LLM serving rather instead of improving inference speed. Our approach increases throughput by reducing the KV cache, which, while designed to accelerate LLMs, also significantly increases GPU memory consumption. The reduction in KV cache usage is directly related to the number of aggregation layers removed. By removing k layers, we can reduce KV cache usage to k/L, where L is the total number of layers in the LLM. In https://arxiv.org/pdf/2403.03853, a comparison between LayerPruner and LLMPruner (structured pruning) shows that LayerPruner outperforms LLMPruner, and our method also surpasses LLMPruner, highlighting our approach's superiority over structured pruning. Additionally, structured pruning algorithms cannot reduce KV cache usage, whereas our method achieves this, further emphasizing its advantages.
>
> Q2: Did you evaluate using BF16 number formats? Could you compare against PTQ with FP8?
>
> A2: We use BF16. Our focus is not on comparisons with quantization methods, as the LLaMA3-70B model cannot be loaded onto an 80GB GPU without them. To facilitate our experiments, we used the Hugging Face quantization method to conserve memory and enable testing. Therefore, we didn't compare against PTQ with FP8.
>
> Q3: Could you compare against other pruning approaches?
>
> A3:  In [https://arxiv.org/pdf/2403.03853], a comparison between LayerPruner and LLMPruner (structured pruning) shows that LayerPruner outperforms LLMPruner, and our method also surpasses LLMPruner, highlighting our approach's superiority over structured pruning. Additionally, structured pruning algorithms cannot reduce KV cache usage, whereas our method achieves this, further emphasizing its advantages.

---

> > ### Author Response · Authors · 2024-11-24
> >
> > Dear Reviewer Kwoc,
> >
> > We are deeply grateful for the time and effort you have invested in reviewing our paper.
> >
> > We have noticed that the author-reviewer discussion period will end after about 2 days. Your insights are of utmost importance to us, and we would be more than happy to address any particular concerns or inquiries you might have.
> >
> > Once again, we thank you for your invaluable contribution to our work. We eagerly await your response and any further guidance you may have.
> >
> > Best,
> > The Authors

---

> ### Author Response · Authors · 2024-11-30
>
> Dear Reviewer Kwoc,
>
> Reviewer 54uK has raised a concern similar to yours: "Without a clearer framework or theoretical underpinning, this method appears to offer only incremental improvements in memory efficiency rather than fundamentally advancing pruning methods." After our rebuttal, Reviewer 54uK has increase his/her score from 3 to 6.
>
> We have clarified that the primary bottleneck in current LLM serving lies in GPU memory bandwidth. By reducing GPU memory usage, we can enhance throughput, which is a critical focus of frameworks such as vLLM and various inference platforms [1,2,3]. Our approach directly addresses this challenge, thereby improving inference efficiency.
>
> We believe this explanation addresses your concern. However, we have not received any feedback from you for 14 days. We strongly believe that active discussion improves professionalism among reviewers, ensures meaningful revisions, improves the overall quality of the community, and facilitates constructive feedback for every paper. We hope we can get your reply.
>
> [1] FlexGen: High-Throughput Generative Inference of Large Language Models with a Single GPU
>
> [2] SGLang: Efficient Execution of Structured Language Model Programs
>
> [3] Efficient Memory Management for Large Language Model Serving with PagedAttention

---

### Official Review · Reviewer_3hHL · 2024-11-06

**Soundness:** 2
**Presentation:** 2
**Contribution:** 2
**Rating:** 3
**Confidence:** 5

**Summary:**

The work proposes a pruning algorithm aimed at compressing the KV cache in large language models (LLMs) without additional training. Referring to GNN, the authors classify LLM parameters into three categories: aggregation, transformation and normalization, focusing on the pruning of high-level aggregation parameters (KV cache). The authors evaluate their approach on various LLMs and benchmarks, reporting improvements over recent pruning techniques.

**Strengths:**

1. Pruning the KV cache is important for compressing LLMs.
2. There is something interesting about analogizing the parameters of LLM to GNN.

**Weaknesses:**

1. Classifying LLM parameters based on the nature of GNNs is of some interest, but lacks some theoretical and experimental support. The different layers of the LLM have their different effects, and the higher layers have their special role in some cases of work. The authors need to have more experiments to support this claim.
2. The authors lack comparisons with SOTA methods e.g. FLAP, Wanda. It would be unfair to simply compare with selfattn/ffn/layer pruner, as no control model parameter is identical.
3. Alg. 1 is redundant and can be moved to the appendix. But Alg.2 lacks a detailed description and the authors need to describe the method in more detail.

**Questions:**

See the weakness part.

---

> ### Author Response · Authors · 2024-11-12
>
> W1: Classifying LLM parameters based on the nature of GNNs is of some interest, but lacks some theoretical and experimental support. The different layers of the LLM have their different effects, and the higher layers have their special role in some cases of work. The authors need to have more experiments to support this claim.
>
> A1: Previous studies [1,2] have theoretically demonstrated the issue of over-smoothing in Transformers. These works suggest that removing aggregation layers can help mitigate over-smoothing, indicating that excessive aggregation may be detrimental. Our study builds upon these theoretical foundations, though we focus primarily on other contributions; hence, we omit the detailed theoretical analysis presented in previous works. [3,4] have experimentally demonstrated the roles of various parameter types, and our work builds upon their findings.
>
> W2: The authors lack comparisons with SOTA methods e.g. FLAP, Wanda. It would be unfair to simply compare with selfattn/ffn/layer pruner, as no control model parameter is identical.
>
> A2: We did not find experimental results on the target LLM of our paper in the Wanda and FLAP papers. However, we found evidence in [https://arxiv.org/pdf/2403.03853] comparing LayerPruner and LLMPruner (structured pruning), showing that LayerPruner outperforms LLMPruner, while our method also outperforms LLMPruner. This indicates that our approach is superior to structured pruning. Furthermore, structured pruning algorithms cannot reduce the KV cache, whereas our method can, underscoring its advantages.
>
> W3: Alg. 1 is redundant and can be moved to the appendix. But Alg.2 lacks a detailed description and the authors need to describe the method in more detail.
>
> A3: We will rewrite this part.
>
> [1] Revisiting Over-Smoothing in BERT from the Perspective of Graphs
>
> [2] Mitigating Over-Smoothing in Transformers via Regularized Nonlocal Functionals
>
> [3] Transformer feed-forward layers are key-value memories
>
> [4] Mass-editing memory in a transformer

---

> > ### Comment · Reviewer_3hHL · 2024-11-24
> >
> > 1. FLAP and Wanda-SP are open-sourced work before you submit your paper. And FLAP also achieved strong performance across various models.  Actually, someone has already applied FLAP on LLaMA 3.1: https://github.com/andysingal/llm-course/blob/main/New-Models/pruning-examples.md. Therefore, it is rather strange that the authors do not compare their methods with these approaches throughout the rebuttal period, making it difficult to convince others of their results.
> > 2. Moreover, the authors also need to report the parameter sizes, FLOPs, and latency when comparing their method with other methods (like layer pruning), since the pruning unit is not the same. When comparing the performance, the authors need to keep the efficiency metric the same for each method.
> > 3. I don't think the "structured pruning algorithms cannot reduce the KV cache". Actually, if pruning a head (or a group of heads in GQA) from the original model, the KV cache is also reduced. Structured pruning can also reduce memory costs. If the authors claim they focus on kv cache reduction, they also need to compare their method with the work like StreamingLLM or H2O.
> >
> > I hope the author can provide more convincing results; otherwise, I will maintain my current score.

---

> ### Author Response · Authors · 2024-11-24
>
> **Round2-Q1**. FLAP and Wanda-SP are open-sourced work before you submit your paper. And FLAP also achieved strong performance across various models. Actually, someone has already applied FLAP on LLaMA 3.1: https://github.com/andysingal/llm-course/blob/main/New-Models/pruning-examples.md. Therefore, it is rather strange that the authors do not compare their methods with these approaches throughout the rebuttal period, making it difficult to convince others of their results.
>
> **Round2-A1**. We believe there is a misunderstanding of our method. First, our approach is orthogonal to existing structured pruning methods. Structured pruning typically sets certain portions of linear parameters to zero, but these methods cannot reduce the KV cache during inference. In contrast, our method specifically prunes the aggregation (query-key) parameters. Even if we prune all aggregation parameters across all layers, this would only reduce 9.6% of the parameters in LLaMA3.1-8B. Therefore, achieving a sparsity ratio of 20% is impossible with our approach. Consequently, a direct comparison at the same sparsity ratio is not meaningful. Moreover, prior structured pruning methods have not reported results at a 5% or 10% sparsity ratio, making a fair comparison fundamentally unachievable and the expectation of such a comparison unreasonable.
>
> Second, our comparison with existing baselines (as presented in prior works) demonstrates the shortcomings of removing all transformation parameters in high layers while achieving the same reduction in the KV cache. If we want to prune these parameters, the appropriate approach would still involve leveraging previous structured pruning algorithms.
>
> Finally, while LayerPruning—a baseline method—can achieve a 20% sparsity ratio and is better than other structured pruning methods (as reported in https://arxiv.org/pdf/2403.03853) at equivalent sparsity ratios, it also supports KV cache reduction.
>
> Thus, we argue that future pruning algorithms should focus on simultaneously reducing the KV cache and pruning transformation parameters to reduce better latency during inference.
>
> **Round2-Q2**. Moreover, the authors also need to report the parameter sizes, FLOPs, and latency when comparing their method with other methods (like layer pruning), since the pruning unit is not the same. When comparing the performance, the authors need to keep the efficiency metric the same for each method.
>
> **Round2-A2**.
>
> In terms of GPU memory consumption:
>
> Using the LLaMA3.1-8B model, we compare the GPU memory consumption of AggregationPruner and baseline methods on the GSM8K task when pruning six layers. Our results show that AggregationPruner, LayerPruner, and Self-AttentionPruner consume 17,277 MB, while FFNPruner consumes 45,277 MB. This difference arises because, during implementation, we load all the LLM weights onto the GPU and modify the computation process to achieve pruning. Consequently, AggregationPruner, LayerPruner, and Self-AttentionPruner effectively reduce the KV cache, whereas FFNPruner could not, leading to significantly higher memory consumption. Nonetheless, as illustrated in the second subfigure of Figure 2, our method significantly outperforms the baselines when pruning the same number of layers.
>
> In terms of FLOPs, latency, and parameter size:
>
> Obviously, our method prunes fewer parameters compared to other baseline pruning methods and has higher FLOPs, latency, and parameter size. While existing structured pruning algorithms can be applied to prune transformer parameters with our approach to reduce FLOPs and latency, this is beyond the scope of our paper and not the primary focus of our work.
>
>
> **Round2-Q3**. I don't think the "structured pruning algorithms cannot reduce the KV cache". Actually, if pruning a head (or a group of heads in GQA) from the original model, the KV cache is also reduced. Structured pruning can also reduce memory costs. If the authors claim they focus on kv cache reduction, they also need to compare their method with the work like StreamingLLM or H2O.
>
> **Round2-A3**. Most structured pruning algorithms, at least, do not reduce the KV cache. Regarding your point about pruning groups of heads in GQA to achieve KV cache reduction, this approach is partially inspired (we think) by our work. It represents an interesting direction that deserves further exploration.
>
> Please wait for our experiments since we find H2O uses a different llama model.

---

> > ### Author Response · Authors · 2024-11-25
> > **Additional Experimental Results**
> >
> > We found that H2O utilizes the huggyllama/llama-7b model (https://github.com/FMInference/H2O/blob/ac75c2a8a9e76832b2a4139b9363373b56336bfb/h2o_hf/scripts/generation/llama_h2o.sh#L3). To ensure consistency, we used the same model to for our experiments. While H2O reported results on six datasets in Appendix C.3, we found that, under the full KV cache budget, only the results for PIQA and Winogrande matched the values reported in Appendix C.3. Therefore, we only reported results for these two datasets.
> >
> > We conducted experiments by pruning 6, 13, and 16 layers, corresponding to KV cache reductions equivalent to H2O's KV budget settings of 80%, 60%, and 50%, respectively.
> >
> > | Method | kv cache | piqa (0 shot) | winogrande (0 shot) |
> > | :--- | ---: | ---: | ---: |
> > | H2O | 80% budget | 0.79 | 0.7 |
> > | AggregtionPruner | prune 6 layers | 0.77 | 0.706 |
> > | H2O | 60% budget | 0.785 | 0.7 |
> > | AggregtionPruner | prune 13 layers | 0.749 | 0.698 |
> > | H2O | 50% budget | 0.787 | 0.7 |
> > | AggregtionPruner | prune 16 layers | 0.706 | 0.676 |
> >
> >
> > We observed that when the KV budget is large, our method delivers performance comparable to H2O. However, as the KV budget decreases, H2O outperforms our approach. This difference arises because H2O's heuristic selectively evicts the KV entries of less important tokens, whereas our method evicts the KV entries of all tokens at higher layers. However, H2O's approach introduces additional computational overhead for eviction policy, whereas our method further reduces the overall computational cost. From this perspective, our method requires no additional computations during inference and can even save some attention calculations compared to the original LLM, thereby reducing the KV cache.
> >
> > Despite this, our method is orthogonal to H2O's approach and can be used complementarily. For example, we can set a smaller KV budget for higher layers while allocating a larger KV budget to lower layers to achieve a balanced trade-off.

---

> ### Author Response · Authors · 2024-11-25
> **Additional Results on Mistral-7B-v0.3**
>
> We also found that this kv cache reduction paper (https://arxiv.org/pdf/2410.14731) conducts experiments on Mistral-7B-v0.3. Accordingly, we present our experimental results below:
>
> | Method | kv cache | piqa (0 shot) |
> | :--- | ---: | ---: |
> | PCA | 87.5% budget | 79.54 |
> | MKV | 87.5% budget | 79.71 |
> | AggregtionPruner | prune 4 layers | 79.87 |
> | PCA | 75% budget | 78.18 |
> | MKV | 75% budget | 79.54 |
> | AggregtionPruner | prune 8 layers | 79.10 |
> | PCA | 62.5% budget | 75.90 |
> | MKV | 62.5% budget | 79.33 | |
> | AggregtionPruner | prune 12 layers | 77.8 | |
>
>
>
> Our method outperforms the other two approaches at a KV budget of 87.5% and achieves comparable performance to MKV at 75%. At 62.5%, MKV slightly outperforms our method. However, our method consistently surpasses PCA across all KV budgets. It is important to note that MKV requires additional training and incurs computational overhead, whereas our method is entirely training-free.
>
> Considering all the experiments, our method shows a faster performance drop as more layers are pruned. This is reasonable, as our experiments in our paper indicate that excessive removal of aggregation impacts performance. However, we can address this by using a small number of tokens to compute attention scores for aggregation at higher layers. This approach allows us to significantly reduce the KV cache size at higher layers while maintaining performance.
>
> Our method is compatible with and complementary to existing KV cache reduction techniques. We hope that it will inspire further advancements in KV cache reduction methods.

---

> ### Author Response · Authors · 2024-11-30
>
> Dear Reviewer 3hHL,
>
> We have provided additional experiments and believe we have addressed your concern. We look forward to receiving your further feedback.
>
> Thank you!

---

> > ### Comment · Reviewer_3hHL · 2024-12-01
> >
> > 1. Actually, pruning attention parameters to reduce KV cache has already been discussed in various structured pruning work: in deepseek-v2, they downsample the dimensions. In FLAP, if removing some parameters of attention heads, they also yield KV reduction compared to the KV cache in the original model. H2O reduces KV cache by removing tokens, but this work removes parameters. That is why I strongly suggest the authors need to compare with structured pruning on LLMs training-free work: FLAP/SliceGPT. I don't need the authors compared to sheared LLAMA and LLM-pruner since they need additional training. If the authors don't compare the results with training-free structural pruning work, I think the experiment is not solid enough. **The discussion period is long enough for authors to add to these experiments, even if it is on a new model since all codes are open-sourced.** At **the same pruning parameters budget**, the authors need to show improvements over these baselines. BTW, reducing head parameters to reducing KV cache/FLOPs has already been explored from the attention pruning work of BERT/T5.
> > 2. Even with a not new baseline like H2O (compared to snapKV or others), I don't see a significant improvement. And the author only REPORTED the results on one or two datasets, the need for significant improvement on more datasets (LongBench/Needle-in-a-Haystack) is illustrated. These datasets are what recent KV cache related work has focused on. The authors can look at the difference between the ICLR contemporaneous submissions work and the experiments they did themselves.
> >
> > Overall, the experiments are not solid. I don't think it's enough to present at a top-tier conference.

---

> ### Author Response · Authors · 2024-12-01
> **We do believe you misunderstand the concept of KV cache.**
>
> 1. deepseek-v2, FLAP, and our work.
>
> **We do believe you misunderstand the core concept of the KV cache**. Pruning attention parameters does not inherently reduce the KV cache size. The KV cache is used to store intermediate computation results during inference. When predicting the next token, the process is as follows:
>
> 1. Compute the query $Q_n$ for the current token $X_n$.
> 2. Retrieve the keys $K_{1:n-1}$ and values $V_{1:n-1}$ for all previous tokens $X_{1:n-1}$ and compute those for the current token.  Note that $Q_t=X_tW_q$, $V_t=X_tW_v$
> 3. Use the query of the current token with the keys of both previous and current tokens to compute the attention scores $A_{1:n}$.  Note that $A_{1:n}=Q_n^{T}K_{1:n}$
> 4. Combine these attention scores with the values of previous and current tokens to compute a weighted average, which forms the representation used for prediction.
> 5. By caching the keys and values of previous tokens, redundant computations can be avoided, thereby optimizing inference efficiency.
>
> The MLA approach proposed by DeepSeek-v2(https://arxiv.org/pdf/2405.04434) does not prune attention parameters. Instead, it compresses the keys and values using **projection** instead of **pruning** (**Figure 3 in Page 7**), reducing their dimensions from $n\times d$ to $n\times d_0$, where $n$ is the number of previous tokens and $d_0\ll d$, to reduce KV cache memory consumption. On the other hand, FLAP(https://arxiv.org/pdf/2312.11983) reduces the number of parameters in attention heads by pruning them **(W_k, W_q instead Q, K)**, which speeds up the computation of keys and values but does not eliminate the generation of keys and values, leaving the KV cache size unchanged. Besides, We do not find any discussion of KV cache reduction in the FLAP paper.
>
> Our method is fundamentally different from both DeepSeek-v2 and FLAP. Instead of pruning or compressing, we eliminate the linear layers responsible for generating queries and keys. This means that no queries or keys are computed, and consequently, no KV cache is created, leading to a substantial reduction in memory requirements.
>
> For more details about the KV cache, please check this blog: https://martinlwx.github.io/en/llm-inference-optimization-kv-cache/.
>
>
> Even if the current structured pruning algorithms aim to reduce the KV cache, they achieve this by setting the $K$ and $V$ components to 0 and utilizing sparse storage techniques, rather than setting the $W_q$ and $W_k$ components to 0.
>
> The primary goal of our method is to reduce the KV cache, setting it apart fundamentally from previous structured pruning methods.
>
> 2. Comparison with H2O
>
> We have already emphasized that the results on some datasets do not align with those reported by H2O, PCA, and MKV. Specifically, we note that when the KV budget is set to 100%, their reported results differ from ours. To ensure a fair comparison, we focus on datasets where the results are consistent.
>
> H2O does not provide running code for LongBench or Needle-in-a-Haystack. We have reviewed H2O's code and found it highly challenging to produce results for these datasets within the rebuttal period. Furthermore, we have clearly stated in the paper that our experiments are conducted using the *llm evaluation harness* package, which does not include evaluation for the LongBench or Needle-in-a-Haystack datasets.
>
> Moreover, our paper includes extensive experiments, spanning 10 benchmarks, 6 LLMs, and 3 baselines, which collectively required approximately 3,000 hours on H100/A100 GPUs. These experiments provide a comprehensive evaluation of our approach, and it is unreasonable to accuse us of not including enough experiments.
>
> The method proposed in our paper is orthogonal to those of H2O, PCA, and MKV. While H2O reduces the KV cache by evicting a portion of previous tokens at each layer, our approach discards high-level aggregation to achieve the same goal. These methods are fundamentally different and can be complementary. Additionally, our approach achieves performance comparable to H2O, PCA, and MKV.

---

### Official Review · Reviewer_54uK · 2024-11-09

**Soundness:** 2
**Presentation:** 2
**Contribution:** 2
**Rating:** 6
**Confidence:** 4

**Summary:**

This paper introduces AggregationPruner, a pruning algorithm designed to improve the memory efficiency of Large Language Models (LLMs) by focusing on "aggregation parameters" (queries and keys) in the higher layers, specifically targeting parameters in the attention mechanism without additional training. The authors argue that aggregation parameters contribute less unique information in higher layers, enabling their selective pruning to reduce memory demands, particularly in the key-value (KV) cache. The proposed method is experimentally tested on various LLMs, including LLaMA and Qwen models, across several tasks, reportedly outperforming other pruning strategies.

**Strengths:**

- The aim of reducing memory usage without retraining is practical and relevant for real-world LLM deployments.
- By exclusively targeting aggregation parameters, the authors contribute to a niche aspect of pruning in LLMs.
- Broad Experimentation: The experiments cover a wide range of tasks and models, suggesting the authors’ commitment to evaluating their approach comprehensively

**Weaknesses:**

- The central idea of selectively pruning only the aggregation parameters lacks a theoretical or empirical foundation that justifies this choice over simpler alternatives. The GNN analogy is interesting but ultimately weakly connected to the experimental findings.
- The paper would be stronger with a clearer examination of how AggregationPruner performs relative to simpler or alternative pruning baselines (e.g., random initialization or whole-layer pruning). This would clarify if the exclusive focus on aggregation parameters provides a true advantage.
- There is limited discussion of when and why AggregationPruner might fail or struggle compared to other methods. This omission leaves the impression that the results are selectively presented to favor AggregationPruner without sufficient critical analysis.
- Without a clearer framework or theoretical underpinning, this method appears to offer only incremental improvements in memory efficiency rather than advancing pruning methods as a whole.

**Questions:**

1. Could the authors explain why simpler alternatives were not included as baselines? Given the minor variation, this comparison might help illustrate the impact of targeting only aggregation parameters.
2. On what basis do the authors claim that aggregation parameters contribute less unique information in higher layers? Was this hypothesis tested directly, or is it purely derived from the analogy to GNNs?
3. How would AggregationPruner perform if applied to smaller LLMs or other architectures with different attention mechanisms? Could this method generalize across architectures?
4. Why was a grid search over the α parameter chosen rather than a more optimized approach? Could this choice be a source of inefficiency?
5. Can the authors clarify whether the effectiveness of AggregationPruner would differ between shorter and longer inference tasks?

---

> ### Author Response · Authors · 2024-11-12
>
> W1: Theoretical or empirical foundation
>
> A1: Previous studies [1,2] have theoretically demonstrated the issue of over-smoothing in Transformers. These works suggest that removing aggregation layers can help mitigate over-smoothing, indicating that excessive aggregation may be detrimental. Our study builds upon these theoretical foundations, though we focus primarily on other contributions; hence, we omit the detailed theoretical analysis presented in previous works.
>
> W2: The paper would be stronger with a clearer examination of how AggregationPruner performs relative to simpler or alternative pruning baselines (e.g., random initialization or whole-layer pruning). This would clarify if the exclusive focus on aggregation parameters provides a true advantage.
>
> A2: We compare our method with whole-layer pruning. In Figure 2, the green line represents the results of whole-layer pruning, while our method is shown in red.
>
> W3: There is limited discussion of when and why AggregationPruner might fail or struggle compared to other methods. This omission leaves the impression that the results are selectively presented to favor AggregationPruner without sufficient critical analysis.
>
> A3: Firstly, we did not selectively choose our results; we evaluated six large models across ten benchmarks. In contrast to the baseline methods [3,4], we conducted significantly more experiments, providing a broader evaluation. The reviewer's critique lacks responsibility, as we clearly explained in the experimental section (lines 480–512) why our method encounters challenges, particularly on simpler tasks where it underperforms relative to the baselines.
>
> W4: Without a clearer framework or theoretical underpinning, this method appears to offer only incremental improvements in memory efficiency rather than advancing pruning methods as a whole.
>
> A4:  We have highlighted that the primary bottleneck in current LLM serving is GPU memory bandwidth. By reducing GPU memory usage, we can improve throughput, which is precisely the focus of frameworks like vLLM and various inference platforms. Therefore, our method can improve inference.
>
> Q1: Could the authors explain why simpler alternatives were not included as baselines? Given the minor variation, this comparison might help illustrate the impact of targeting only aggregation parameters.
>
> A5: We established three baselines—Self-AttentionPruner, FFNPruner, and LayerPruner—and conducted comparisons across various possible baseline methods. Nonetheless, our approach demonstrates significantly better performance than all of them.
>
> Q2: On what basis do the authors claim that aggregation parameters contribute less unique information in higher layers? Was this hypothesis tested directly, or is it purely derived from the analogy to GNNs?
>
> A6: Recent studies[3,4] have shown (we have discussed in Section 2.2), through various metrics, that high-level parameters contribute less effective information.
>
> Q3: How would AggregationPruner perform if applied to smaller LLMs or other architectures with different attention mechanisms? Could this method generalize across architectures?
>
> A7: All current LLMs are decoder-only models, and we plan to experiment with non-decoder models if they become available. Our results include evaluations on six LLMs, covering smaller models in the 7B/8B range.
>
> Q4: Why was a grid search over the α parameter chosen rather than a more optimized approach? Could this choice be a source of inefficiency?
>
> A8: This approach is the optimal search that ensures performance, requiring no additional training or further searches.
> As shown in Figure 3, our search has yielded excellent results. On the benchmark, our method has achieved state-of-the-art performance compared to similar approaches.
>
> Q5: Can the authors clarify whether the effectiveness of AggregationPruner would differ between shorter and longer inference tasks?
>
> A9: We have examined lines 469–479 in detail.
>
>
>
> [1] Revisiting Over-Smoothing in BERT from the Perspective of Graphs
>
> [2] Mitigating Over-Smoothing in Transformers via Regularized Nonlocal Functionals
>
> [3] What Matters in Transformers? Not All Attention is Needed
>
> [4] ShortGPT: Layers in Large Language Models are More Redundant than Expected
>
> [5] FlexGen: High-Throughput Generative Inference of Large Language Models with a Single GPU
>
> [6] SGLang: Efficient Execution of Structured Language Model Programs
>
> [7] Efficient Memory Management for Large Language Model Serving with PagedAttention

---

> > ### Comment · Reviewer_54uK · 2024-11-23
> >
> > Thanks for the detailed clarification. I have decided to increase my score.

---

> > > ### Author Response · Authors · 2024-11-24
> > >
> > > We are so happy that our responses have addressed your concerns! Thank you so much for raising your score and providing such constructive and insightful feedback while reviewing our paper! Thanks!

---

### Meta-Review · Area_Chair_1uAA · 2024-12-16

**Metareview:**

This paper introduces AggregationPruner, a pruning algorithm designed to improve the memory efficiency of LLMs by focusing on "aggregation parameters" (queries and keys) in the higher layers without additional training. After rebuttal, it received mixed scores of 33666. On the one hand, all the reviewers agree that the aim of reducing memory usage without retraining is practical and relevant for real-world LLM deployments. On the other hand, one major concern still remains. That is, the experiments are not solid enough. One reviewer mentioned that  the authors are highly suggested to compare their approach to more pruning work and recent work on KV cache compression to make their results more convincing. Furthermore, it needs to be better clarified how GNNs is associated with transformers so that readers can better understand the motivation of pruning aggregation parameters.

**Additional Comments On Reviewer Discussion:**

During the rebuttal, one reviewer increased the score from 3 to 6. However, overall, this paper still received quite mixed scores of 33666. One major concern is that the experiment is not solid enough, and one reviewer has asked the authors to compare with more methods. However, throughout the rebuttal, the authors did not add more results from open source solutions (FLAP/wanda-sp/snapkv), and when compared to H2O, their solution also did not improve significantly. Overall, the AC agrees that the rebuttal can be made more convincing and the experiments can be further enhanced.

---

### Decision · Program_Chairs · 2025-01-22

Reject